

# A thicker, rather than thinner, East Antarctic Ice Sheet plateau during the Last Glacial Maximum

Cari Rand[1], Richard S. Jones[1], Andrew N. Mackintosh[1], Brent Goehring[2], Kat Lilly[3]

[1]Securing Antarctica's Environmental Future, School of Earth, Atmosphere and Environment, Monash University, Wellington Road, Clayton, Melbourne, Victoria 3800, Australia

[2]Los Alamos National Laboratory, Bikini Atoll Road, Los Alamos, New Mexico 87545, USA

[3]RSC, PO Box 5647, Dunedin, New Zealand

*Correspondence to*: Cari Rand (cari.rand@monash.edu)

**Abstract.** In this study, we present a surface-exposure chronology of past ice-thickness change derived from *in-situ* cosmogenic-$^{14}$C dating at a site on the edge of the East Antarctic plateau, 380 km inland from the Antarctic coastline. Our knowledge of how the Antarctic ice sheet has responded to Quaternary climate change relies on a combination of geological data and ice-sheet modeling. At the Last Glacial Maximum (LGM), observations and models suggest that increased ice-sheet volume was accommodated by thicker ice near the coast and grounding-line advance towards the continental-shelf edge. In contrast, the ice sheet interior maintained a relatively stable thickness until present, with ice-core evidence even suggesting thinner ice relative to today. However, the magnitude of these thickness changes, and the location dividing thicker versus thinner ice at the LGM is poorly constrained. Geological reconstructions of past ice thickness in Antarctica mostly come from surface-exposure data using cosmogenic nuclides that are relatively insensitive records of ice-cover changes on timescales of tens of thousands of years. This can lead to inaccurate records of LGM ice thickness, particularly towards the East Antarctic plateau, where cold-based non-erosive ice may inhibit bedrock erosion. Samples saturated with $^{14}$C at 1912 m a.s.l. indicate that the summit of Nunatak 1921 was exposed during the LGM, while unsaturated samples indicate that thinning subsequently occurred, with some (25-45%) post-LGM thinning recorded at ~15-11 ka and most (55-75%) recorded during the Holocene. These results imply that at least part of the interior East Antarctic Ice Sheet (EAIS) was thicker at the LGM than it is now, and that gradual ice-sheet thinning began ~15 ka. Ice-sheet models that do not account for this thickness change would inaccurately characterize the LGM geometry of the EAIS and underestimate its contributions to deglacial sea-level rise.

## 1 Introduction

The East Antarctic Ice Sheet (EAIS) is the largest contiguous mass of ice on Earth (Rignot *et al*., 2019). Loss of ice to melting and calving is predicted to be offset by increases in snow accumulation over the coming century, but beyond 2100 CE, the ice sheet is expected to lose mass and contribute to sea-level rise (Stokes *et al*., 2022). Characterizing past changes of the EAIS is necessary for several reasons:

1. Satellite observations of Antarctic glaciers extend back only to the 1960s, so other records of past ice-sheet states are needed in order to reliably distinguish long-term trends from natural variability (Hanna *et al*., 2020; Jones *et al*., 2022);



2.  Geodetic data used to estimate modern ice-mass changes must be corrected for glacial isostatic adjustment (e.g., Coulon *et al.*, 2021), the magnitude of which is dependent on the past configuration of the ice sheet;

3.  Determining the magnitude and timing of ice loss can identify or exclude potential sources of meltwater input to oceans during past periods of rapid sea-level rise (e.g., Lin *et al.*, 2021); and

4.  Numerical models informed by records of past ice-sheet change are used to estimate the future contributions to sea-level rise (e.g., DeConto *et al.*, 2021).

However, reconstructing the geometry of the EAIS is challenging. Evidence of past ice thickness comes from radar, ice-core and geological data, which are sparse owing to the remoteness of East Antarctica, the large area of the ice sheet, and the sparsity of ice-free areas. Furthermore, different records of LGM ice thickness are often in disagreement with one another.

During the Last Glacial Maximum (LGM), at approximately 20 ka, available evidence points towards a more extensive but shallower-gradient ice sheet (Mackintosh *et al*., 2014). Dated acid-insoluble organic matter in sediments from the East Antarctic coast indicate that the EAIS advanced to the edge of the continental shelf in most locations during the LGM (Bentley *et al.*, 2014), with constraints from cosmogenic $^{10}$Be and $^{26}$Al indicating the presence of ice near the coast that was thicker than it is now (e.g., Mackintosh *et al*., 2007; White *et al*., 2011; Yamane *et al*., 2011). Meanwhile, snow-accumulation rates interpolated between ice domes indicate a thinner ice sheet across the East Antarctic plateau (Buizert *et al*., 2021) at the LGM. A "hinge zone" thus likely existed between thicker ice at the coast and thinner ice in the interior during the LGM relative to today (Bockheim *et al*., 1989; Andersen *et al*., 2023), but the location of this transition point across East Antarctica is unclear. Cosmogenic $^{10}$Be and $^{26}$Al ages from ice-free areas on the edge of the East Antarctic plateau such as the Grove Mountains or southern Prince Charles Mountains are older than the LGM (e.g., Lilly *et al*., 2010; White *et al*., 2011), implying no change since or slightly thinner ice in these locations at the LGM (**Fig. 1**).

Yet existing cosmogenic-nuclide data from regions of cold-based non-erosive ice may not provide reliable constraints on LGM ice thickness. Many samples have apparently pre-LGM and inconsistent $^{10}$Be and $^{26}$Al exposure ages, indicating nuclides inherited from previous periods of exposure (Balco *et al*., 2014). Due to the short half-life of *in situ* $^{14}$C (5.7 kyr), its concentration decays quickly when shielded (e.g., when covered by ice; Goehring, Balco, *et al*., 2019); this makes *in situ* $^{14}$C a useful tool for investigating post-LGM glacial history (Nichols, 2022).



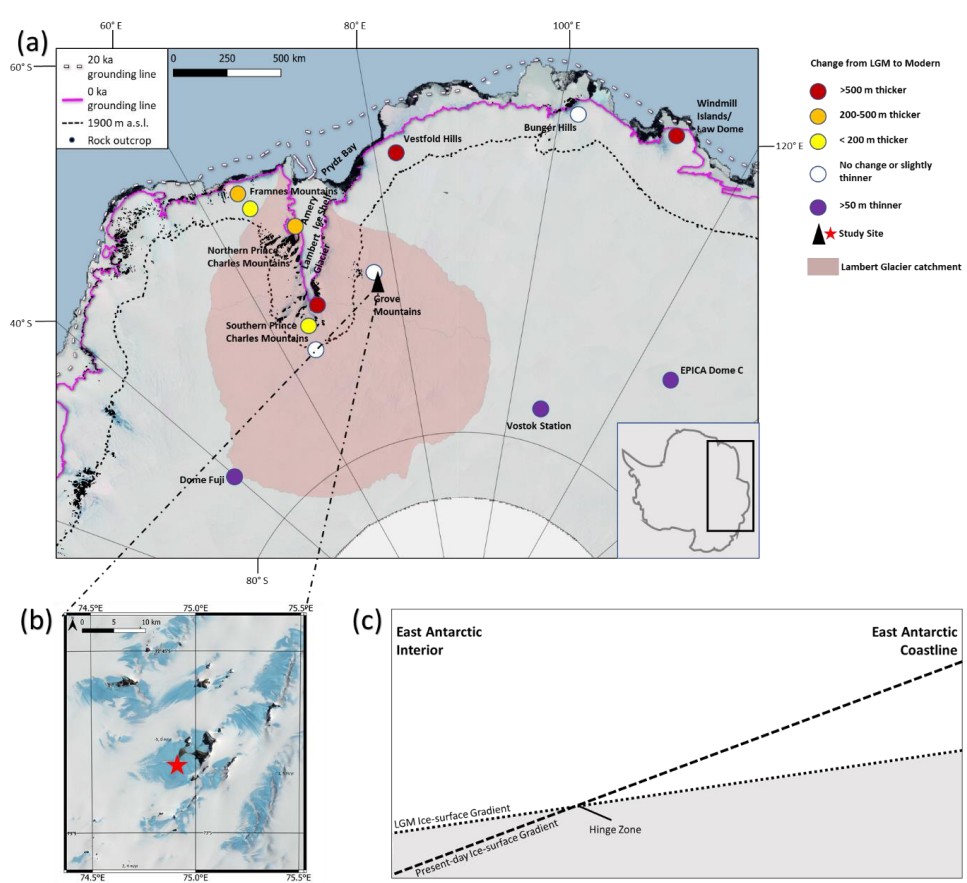

**Figure 1: Constraints on central East Antarctic ice thickness at the Last Glacial Maximum (LGM). (a)
Inferred LGM-to-present ice-thickness differences near Lambert Glacier. Dashed black line shows the 1900
m a.s.l. contour (Liu *et al.* 2015), the elevation of our sampled nunatak. This line represents the most interior
geological evidence and reflects a potential hinge zone between coastal and interior LGM-ice-thickness
change. Elements of this map were provided by the Quantarctica 3 GIS package provided by the Norwegian
Polar Institute (Matsuoka *et al.*, 2018), including ice-free areas (Burton-Johnson *et al.*; 2016), the current
Antarctic-ice-sheet grounding line (Bindschadler *et al.*, 2011), and the inferred East Antarctic grounding
line 20 ka (Bentley *et al.*, 2014). Red-shaded ice indicates the extent of the catchment of Lambert Glacier
(Zwally *et al.*, 2012). LGM thickness data for this figure come from Buizert *et al.* (2021; Dome Fuji and
EPICA Dome C), Lilly *et al.* (2010; Grove Mountains), Mackintosh *et al.* (2007; Framnes Mountains),
Mackintosh *et al.* (2014) and references therein (Bunger Hills, Law Dome, Vestfold Hills, Vostok Station,
and Windmill Islands), and White *et al.* (2011; Prince Charles Mountains). (b) Satellite view of the study
area with the sampled nunatak (Nunatak 1921). Bedrock and erratic samples were collected in a transect
extending from the modern ice surface to nunatak summit. (c) Diagram illustrating the concept of a "hinge
zone" in ice-thickness change. Image shows hypothetical vertically exaggerated cross-sections of the East
Antarctic Ice Sheet at the LGM (dotted line) and present day (dashed line). If coastal ice thins and interior**





**82** **ice thickens after the LGM, the modern ice-surface profile would intersect the LGM surface profile**

**83** **somewhere in the middle; this intersection is the "hinge zone", at which ice there has been no net change in**

**84** **ice thickness since the LGM. LGM-thickness reconstructions in White *et al*. (2011) and Lilly *et al*. (2010)**

**85** **placed the "hinge zone" in areas equivalent to a present-day ice-surface elevation of ~1900 m a.s.l.**

**86**

**87** In this study, we aim to constrain how far inland the EAIS was thicker at the LGM than it is at present by testing

**88** previously-measured samples at a key site in the ice sheet interior using *in situ* $^{14}$C. Rocks exposed since before

**89** the LGM should be saturated with $^{14}$C, meaning that the rates of *in situ* $^{14}$C production and decay are equal (a state

**90** which requires 3-5 half-lives of exposure to reach; Dunai, 2010). Conversely, exposed rocks with less-than-

**91** saturated concentrations of $^{14}$C from a site in East Antarctica imply that those samples were covered at some time

**92** since the LGM by a thicker-than-present EAIS that subsequently thinned. The concentration of a cosmogenic

**93** nuclide in a sample will remain at secular equilibrium indefinitely unless disturbed by cover, erosion, or transport;

**94** thus, only a minimum age can be assigned to saturated samples. Measuring samples from an elevation transect

**95** with *in situ* $^{14}$C thus allows us to reevaluate the ice-thickness history at the site: the ice must have been thick

**96** enough to cover at least the highest-elevation unsaturated sample, and had to have been that thick within the time

**97** it would have taken for the $^{14}$C concentration of that sample to reach saturation again.

**98** **1.1 Study Area**

**99** The Grove Mountains are well located to assess how far inland the EAIS was thicker at the LGM than it is at

**100** present and whether previously measured concentrations of $^{10}$Be and $^{26}$Al from this site likely reflect a component

**101** of nuclides inherited from a previous period of exposure. These isolated nunataks are located ~200 km upstream

**102** of the main trunk of Lambert Glacier and ~400 km inland/south of the Antarctic coast (**Fig. 1**) and are the most

**103** interior ice-free area in this region. The summits of the nunataks rise 100-200 m above the modern ice surface

**104** (~1800 m a.s.l.), providing the potential to record past EAIS-thickness changes. Ice flows slowly (<5 m yr$^{-1}$) to

**105** the west-northwest between these nunataks (Rignot *et al*., 2011).

**106** At Nunatak 1921, evidence of past ice cover is apparent from the occurrence of felsic cobbles atop very weathered

**107** orthogneiss bedrock (Lilly, 2008). Given the sparsity of outcrops and non-channelized nature of ice flow in the

**108** interior EAIS, we are not able to identify the provenance of these cobbles beyond stating that they are not locally

**109** derived (i.e. they are erratics).

**110** **2 Methods**

**111** Here we reanalyze samples first presented in Lilly *et al*. (2010), which were collected from the Grove Mountains

**112** for $^{10}$Be and $^{26}$Al analysis as part of a study of the long-term glacial history of the region. Measurements of $^{10}$Be

**113** and $^{26}$Al were carried out in 2004 at the ANTARES Accelerator Mass Spectrometry facility. Nuclide

**114** concentrations below saturation were recorded for all samples, indicating 40-700 kyr of exposure since the

**115** bedrock was last reset. For full details, see Lilly (2008) and Lilly et al. (2010).

**116** The samples were collected in an elevation transect from the present-day ice surface on the upstream face of

**117** Nunatak 1921 in 2003/4 and 2004/5 (Lilly *et al*., 2010; **Table 1**). Pairs of bedrock and erratic samples showed no

**118** evidence of post-depositional movement, cover by sediments, or subaerial erosion. Samples were preferentially



collected from ridgetops to minimize the chances of shielding by snow. As neither plucking scars nor glacial striae
were observed at the site (Lilly *et al*., 2010), indicating low or negligible rates of subglacial erosion, we anticipate
that the existing $^{10}$Be and $^{26}$Al concentrations do not accurately record LGM ice thickness.
To provide a test of LGM ice thickness, we carried out *in situ* $^{14}$C analysis on ten of these samples that form a
transect covering 96 m of elevation (1825-1921 m a.s.l.). Seven of the samples (GR01, GR03, GR04, GR06,
GR07, GR13, and GR18) were erratic cobbles. The remaining three (GR12, GR15, and GR21) were bedrock
samples.




**Table 1: Sample locations**

| Sample ID | Elevation (m a.s.l.) | Elevation above modern ice surface (m) | Latitude (degrees S) | Longitude (degrees E) | Thickness (cm) | Topographic shielding | Lithology |
|---|---|---|---|---|---|---|---|
| GR01 | 1832 | 7 | 72.9115 | 74.9096 | 2 | 0.985 | Felsic metamorphic |
| GR03 | 1854 | 29 | 72.9115 | 74.9079 | 2 | 0.992 | Quartzite |
| GR04 | 1870 | 45 | 72.9110 | 74.9067 | 2 | 1.000 | Quartzite |
| GR06 | 1894 | 69 | 72.9099 | 74.9044 | 2 | 0.998 | Fine-grained felsic |
| GR07 | 1921 | 96 | 72.9088 | 74.9045 | 2 | 1.000 | Quartzite |
| GR12* | 1825 | 0 | 72.9112 | 74.9097 | 2 | 0.985 | Orthogneiss |
| GR13 | 1839 | 14 | 72.9115 | 74.9094 | 2 | 0.993 | Unknown |
| GR15* | 1847 | 22 | 72.9115 | 74.9088 | 3 | 0.993 | Orthogneiss |
| GR18 | 1873 | 48 | 72.9108 | 74.9061 | 4 | 1.000 | Vein quartz |
| GR21* | 1912 | 87 | 72.9088 | 74.9045 | 3 | 0.999 | Orthogneiss |

**A density of 2.7 g cm⁻³ is assumed for all samples. Bedrock-sample IDs are marked with an asterisk; all**
**other samples were erratic cobbles.**

Quartz was isolated through physical and chemical processing at the Tulane University Cosmogenic Nuclides
Laboratory (TUCNL; Goehring *et al*., 2019). Whole samples were crushed and milled, then all samples were
sieved to select their 125-500-micron fractions. Sieved samples were then rinsed with tap water to remove clay-
sized grains. A roller-type magnetic separator was then used to remove magnetic minerals. Froth flotation was
used to separate quartz and feldspar grains, followed by etching for at least two days in 5% HF/HNO₃ on a shaker
table and at least two days in a sonicator in 1% HF/HNO₃ in order to remove adsorbed carbon species (Nichols
and Goehring, 2019).
Following the isolation and purification, 0.6-5 g aliquots were separated from the cleaned quartz for ¹⁴C extraction.
Before extraction, each aliquot was sonicated in 50% HNO₃ for 0.5 hr, then rinsed with Type I water and dried
overnight in a vacuum oven. The dried quartz was then loaded into a LiBO₂-flux-containing Pt crucible and step
heated in O₂ for 0.5 hr at 500 °C and 3 hr at 1,100 °C in the Tulane University Carbon Extraction and
Graphitization System. Carbon species released were oxidized to CO₂ over hot quartz, then cryogenically purified,
collected, and diluted with ¹⁴C-free CO₂. An aliquot of this gas was separated for δ¹³C analysis and the remainder
graphitized via Fe-catalyzed H₂ reduction. For further details, see the method of Goehring *et al*. (2019).
Concentrations of ¹⁴C were then measured at the National Ocean Sciences Accelerator Mass Spectrometry facility
at the Woods Hole Oceanographic Institution, and data reduction followed Hippe and Lifton (2014).
A blank value of 58,000 ± 3,110 atoms was subtracted from the total measured atoms from each sample; this value
represents the continually updated mean value of process blanks run at the TUCNL since April, 2016 (Goehring
*et al*., 2019). This blank-corrected measurement was divided by the run mass to determine the ¹⁴C concentration
of each sample. Exposure ages were calculated using the "LSDn" nuclide-specific production rate scaling scheme
of Lifton *et al*. (2014). The production rate of *in-situ* ¹⁴C was calibrated using the CRONUS-A interlaboratory
comparison material (Goehring *et al*., 2019). The CRONUS-A material is assumed to be saturated with *in-situ* ¹⁴C
based on geological observations indicating that its collection site has not been covered in the last 11.3 Myr
(Goehring *et al*., 2019; Nichols *et al*., 2019). Repeated measurements of CRONUS-A material at the TUCNL



show ~6% variation in $^{14}$C concentrations; thus, we use a minimum uncertainty equal to 6% of the calculated $^{14}$C
concentration of our samples for exposure-age calculation (**Table 2**).

**Table 2: Sample $^{14}$C concentrations and exposure ages**

| Sample number | $[^{14}C]$ ($10^5$ atoms g$^{-1}$) | $^{14}$C Age (ka) | Internal $^{14}$C-age uncertainty (ka) | External $^{14}$C-age uncertainty (ka) |
|---|---|---|---|---|
| GR01 | $0.86 \pm 0.08$ | 1.021 | 0.098 | 0.100 |
| GR03 | $2.80 \pm 0.17$ | 4.023 | 0.312 | 0.324 |
| GR04 | $5.41 \pm 0.33$ | 11.004 | 1.387 | 1.442 |
| GR06 | $6.20 \pm 0.38$ | 14.660 | 2.452 | 2.549 |
| GR07 | $8.05 \pm 0.49$ | Saturated | N/A | N/A |
| GR12(BR) | $1.14 \pm 0.07$ | 1.393 | 0.091 | 0.095 |
| GR13 | $1.59 \pm 0.10$ | 2.005 | 0.136 | 0.142 |
| GR15(BR) | $0.15 \pm 0.01$ | 0.019 | 0.013 | 0.013 |
| GR18 | $5.47 \pm 0.33$ | 11.600 | 1.530 | 1.590 |
| GR21(BR) | $7.78 \pm 0.11$ | Saturated | N/A | N/A |

**All measurements of $^{14}$C atoms per sample corrected by subtracting a $0.58 \pm 0.31$ atom blank prior to**
**concentration calculation. "Internal" $^{14}$C-age uncertainty includes only instrumental uncertainty.**
**"External" $^{14}$C-age uncertainty includes both instrumental and production-rate uncertainties.**
**3 Results**
Our samples have $^{14}$C concentrations between $15 \pm 0.96$ x $10^3$ atoms g$^{-1}$ (GR15) and $805 \pm 48.3$ x $10^3$ atoms g$^{-1}$
(GR07). The sample with the lowest concentration has an exposure age of $0.02 \pm 0.01$ ka, and the samples with
the highest concentrations are saturated (**Table 2**). These exposure ages are $40 \pm 9$ (GR01) to $262 \pm 22$ kyr (GR03)
less than $^{10}$Be and $^{26}$Al concentrations from each sample (for full sample-measurement details, see **Table S1**).
Samples form a thinning transect with concentrations and ages mostly increasing monotonically with elevation
(**Fig. 2**). There are however two exceptions, both low-elevation bedrock samples (GR15 and GR12). We suspect
that the sites of these two samples may have been covered by snow, other sediment, or a boulder that moved
within the last millennium.



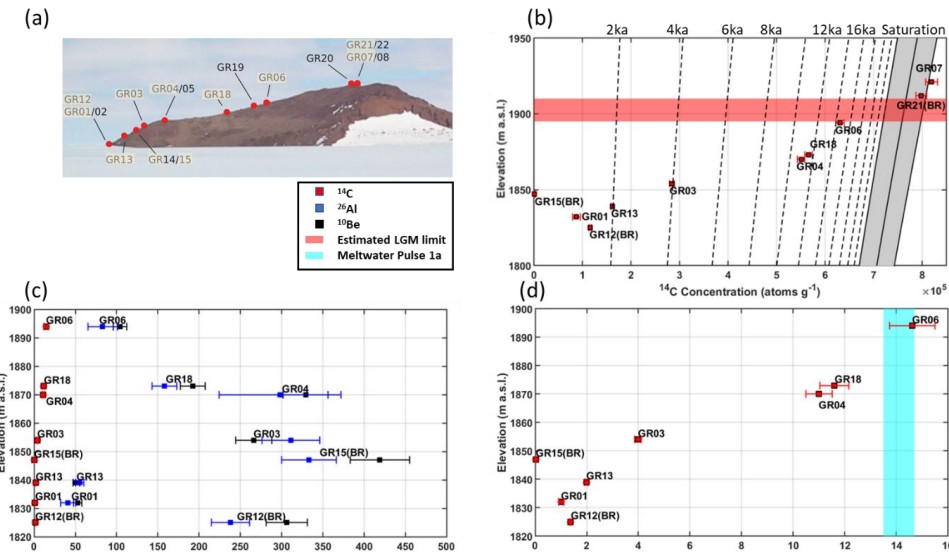


**Figure 2: Sample nuclide concentrations and exposure ages. The current ice surface at this site is roughly**
**coincident with the elevation of sample GR12. (a) Locations of samples noted on a photograph of the south**
**face of Nunatak 1921. IDs of samples from which $^{14}$C was measured in this study are highlighted. This**
**image modified from Lilly *et al*. (2010). (b) $^{14}$C concentrations plotted against elevation. Isochrons (dotted**
**lines) show corresponding exposure ages at each elevation. Tilted vertical gray band to right represents the**
**saturation error envelope. Samples GR07 and GR21 are saturated, indicating >25 kyr of exposure and**
**implying the summit of the nunatak was exposed during the LGM. Horizontal red band indicates the range**
**of possible LGM ice-surface elevations limited by the elevations of GR06, the highest-elevation unsaturated**
**sample, and GR21, the lowest-elevation saturated sample. Samples GR15 and GR21 are out of stratigraphic**
**order and considered outliers. (c) Sample exposure ages plotted against elevation, calculated from**
**concentrations of $^{14}$C (this work) and $^{26}$Al and $^{10}$Be (Lilly *et al*., 2010). Note the younger exposure ages**
**calculated from $^{14}$C. (d) As plot (c), but only showing $^{14}$C exposure ages for the last 16 ka. Light blue bar**
**indicates the timing of meltwater pulse 1a (Deschamps *et al*., 2012). Samples GR07 and GR21 are saturated**
**with $^{14}$C and thus omitted from this plot. See the Supplementary tables for all sample information, nuclide**
**concentrations and calculated exposure ages.**

If the ice-sheet thickness was similar to or thinner than at present in the vicinity of the Grove Mountains at the
LGM, our samples would be saturated with $^{14}$C. However, our samples show a clear trend of increasing $^{14}$C
concentrations with elevation (**Fig. 2**). Only two samples (GR07 and GR21, **Table 2**) show evidence of saturation,
both near the summit of the nunatak. These results thus show that ice was thicker at the LGM than at present in
the Grove Mountains but not sufficiently thick as to override the summits (at least neither lengthily nor deeply
enough to allow nuclide concentrations in these samples to decay below saturation).



The LGM ice surface must have been between the lowest of our saturated and highest of our unsaturated samples,
corresponding to an elevation between 1894 and 1912 m a.s.l. This equates to ice 63-87 m thicker at the LGM
than at present, with subsequent thinning.
Additionally, exposure ages calculated from $^{14}$C concentrations allow us to infer a simple thinning history at
Nunatak 1921. The highest unsaturated sample (GR06) provides a minimum post-LGM age for the onset of
thinning at the site of 14.9 ± 1.0 ka (**Fig. 2, Table 2**). Up to 18 m (21-29%) of thinning could have occurred before
and up to 21 m (24-33%) during meltwater pulse 1a (MWP-1a; **Fig. 2(b)**) assuming a linear thinning history, but
the potential for glacial overshoot, whereby the glacier thins beyond its new equilibrium thickness and
subsequently rethickens, makes these minimum estimates. Most post-LGM thinning (55-70%) is recorded during
the Holocene (the last 11.7 ka; Walker *et al*., 2009). Based on our lowest-elevation sample (GR12), which was
collected less than 1 m above the current ice surface (~1820 m a.s.l.), the present-day ice thickness was reached
at 1.4 ± 0.1 ka (**Tables 1 & 2**).
**4 Discussion**
New exposure ages calculated from *in situ* $^{14}$C concentrations allow us to revise the history of the EAIS at this
site. The combination of saturated and unsaturated samples in the Grove Mountains shows that the highest peaks
were exposed during the LGM, yet the ice sheet was modestly thicker (up to 87 m) here at the LGM than at
present, contrary to previous ice-thickness data at this site and reconstructions of the interior EAIS at the LGM
(e.g. Lilly *et al*., 2010; Buizert *et al*., 2021).
Longer-lived nuclides from our samples ($^{10}$Be and $^{26}$Al) do not show saturation (Lilly *et al*., 2010) but the high
contribution of inherited nuclides from pre-LGM exposure prevents an accurate test of the LGM ice thickness and
reconstruction of the post-LGM thinning history. Our $^{14}$C data indicate that ice cover occurred at this site and the
period of cover was long enough to allow $^{14}$C concentrations in our samples to decay. The summit of the nunatak
was either uncovered or only covered briefly or shallowly (≲10 m) enough for the two summit samples to become
re-saturated with $^{14}$C during the Holocene (**Fig. 3**). Following the LGM, the flanks of the nunatak were re-exposed,
and thinning progressed through to the Late Holocene.





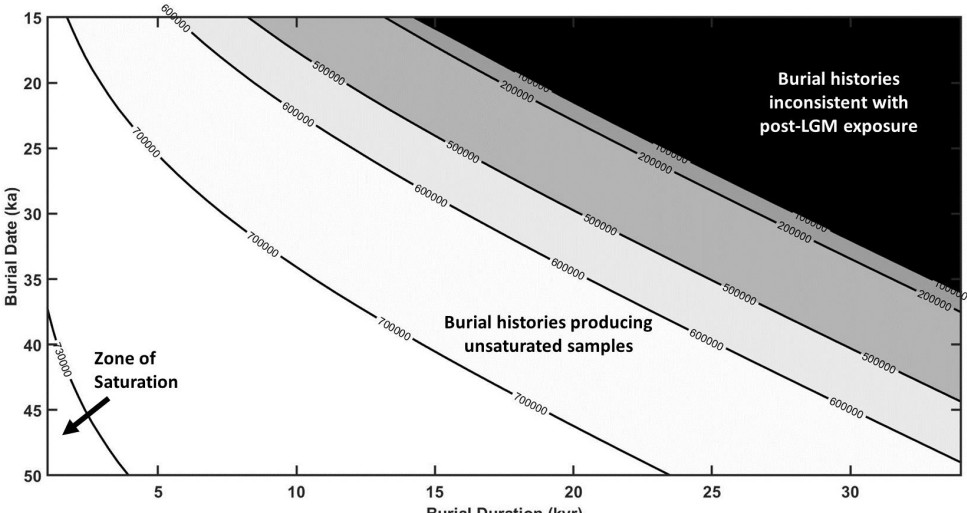


**Figure 3: Burial-history contour plot. Contours show [14]C concentrations resulting from glacial histories with one episode of burial. For legibility, only concentrations corresponding to saturation and sample concentrations measured in this study are shown, and concentrations are rounded down to the nearest $10^5$ atoms g[-1] ($10^4$ atoms g[-1] in the case of the saturation contour). Sample contour affiliations are as follows: GR15 and GR01: 0 atoms g[-1]; GR12 and GR13: $1 \times 10^5$ atoms g[-1]; GR03: $2 \times 10^5$ atoms g[-1]; GR04 and GR18: $5 \times 10^5$ atoms g[-1]; GR06: $6 \times 10^5$ atoms g[-1]; GR21: $7 \times 10^5$ atoms g[-1]. For precise sample concentrations, see Table 2. The black-shaded part of the graph shows histories that require glacial burial after 15 ka, inconsistent with the 14.9 ka exposure of GR06. The grey-shaded part of the graph shows histories that would result in sample GR21 being unsaturated with [14]C. The unshaded portion of the graph shows the uncertainty window of a saturated sample at this latitude and elevation (72.9088°S, 1,912 m a.s.l.; see Table 1). Only the lesser end of the saturation window is consistent with any significant degree of burial under enough ice to effectively stop production (~10 m). Sample GR21 plots off the bottom-left corner of this figure; its [14]C concentration (7.78E+05 atoms g[-1], see Table 1) is inconsistent with any episode of burial longer than 1 kyr in the last 50 ka, indicating constant exposure since the LGM.**

237

Direct constraints from cosmogenic [10]Be and [26]Al show evidence of the ice being thicker near the Antarctic coast at the LGM than at present (e.g., Mackintosh *et al*., 2007; White *et al*., 2011), but exposure ages derived from the same nuclides from interior sites such as the Grove Mountains pre-date the LGM (Lilly *et al*., 2010). While we cannot rule out the thicker-than-present ice at the Grove Mountains being an entirely localized phenomenon, we suggest based on the application of [14]C in this study and other Antarctic studies (e.g., Nichols *et al*., 2019; White *et al*., 2011; Fogwill *et al*., 2014; Hillebrand *et al*., 2021) that at least some previous reconstructions of LGM ice thickness based on longer-lived nuclides (e.g. [10]Be and [26]Al) away from the coast and fastest-flowing parts of East Antarctica may be inaccurate.

Our new chronology indicates that records of thicker-than-present ice near the East Antarctic coast at the LGM may be representative of ice thicknesses further into the interior than previous reconstructions suggest





(Mackintosh *et al.*, 2014). Cosmogenic dating and geomorphological evidence from elsewhere in the Lambert
Glacier catchment support a low-angle ice stream surface at the LGM, with ice 160 m thicker at the most upstream
site in the Prince Charles Mountains (Mt. Ruker), and at least 250 m and up to 800 m thicker at sites closer to the
coast (White *et al.*, 2011; **Fig. 1**). The "hinge zone" between interior and coastal change, where the LGM ice
thickness was the same as today, was proposed to be at ~1900-2000 m a.s.l. based on the available evidence at
the Prince Charles Mountains and Grove Mountains (Mackintosh *et al.*, 2014). A thicker-than-present EAIS at
the Grove Mountains during the LGM therefore indicates that this "hinge zone" lies further inland, increasing the
amount of LGM ice volume across much of the ice sheet (**Fig. 4**).

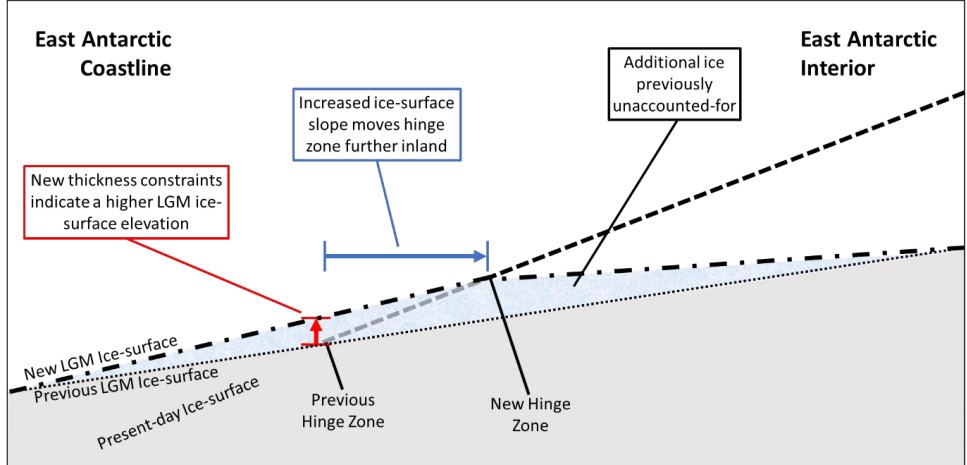

**Figure 4: Implications of new LGM ice thickness constraints on the East Antarctic "hinge zone". Modified**
**from Fig. 1(c), the diagram shows hypothetical vertically exaggerated cross-sections of the East Antarctic**
**Ice Sheet at the present day (dashed line), and at the LGM based on previous evidence (dotted line) and**
**accounting for our data (dot-dashed line). Our results indicate that ice at the Grove Mountains (near the**
**approximate elevation previously considered the "hinge zone") was ~70 m thicker than it is today.**
**Assuming that LGM ice-thickness estimates near the coast are accurate, this necessitates a steeper coastal**
**ice-surface slope to accommodate the increased thickness at the Grove Mountains (and a shallower East**
**Antarctic plateau ice-surface slope if the LGM ice-thickness estimates in the interior are accurate), moving**
**the "hinge zone" further into the interior. The exact gradients of these slopes and location of the "hinge**
**zone" control the volume of ice lost from the East Antarctic Ice Sheet since the LGM.**
An implication of this interior portion of the EAIS being thicker, rather than thinner, at the LGM is that the ice
subsequently thinned, allowing us to evaluate deglacial leads and lags between the coast and interior. The earliest
deglaciation constraints in this region come from ice-sheet thinning in the Prince Charles Mountains at 18 ka
(White *et al.*, 2011; Bentley *et al.*, 2014), which was possibly coincident with grounding-line retreat on the
continental shelf in Prydz Bay (Mackintosh *et al*., 2014). Ice-shelf retreat began by ~16 ka and ~14 ka in west-
central and eastern Prydz Bay, respectively, with the Rauer Group and Vestfold Hills ice-free by ~11 ka (White
*et al.*, 2022). Our record of initial ice thinning in the Grove Mountains at ~15 ka indicates that thinning occurred
~3 kyr earlier in the Prince Charles Mountains, though the timing at the Grove Mountains is broadly consistent



with available evidence of deglaciation at the coast. The modern ice-surface elevation was reached by 9-12 ka at
the Prince Charles Mountains (White *et al.*, 2011) but 1.4 ka in the Grove Mountains, ~7.6-10.6 ka later.
Deglaciation thus possibly started and likely finished earlier downstream, and the magnitude of thinning was
greater at the Antarctic coastline than in its interior. Ice-sheet modeling indicates that responses to sea-level rise,
decreased accumulation, and changes in temperature should manifest first at the margins of the ice sheet, causing
thinning to propagate into the interior of the ice sheet (Alley and Whillans, 1984; Spector *et al*., 2019). Such
propagation is likely slowed and attenuated by distance and travel over bedrock highs (Johnson *et al*., 2021), such
as the Grove Mountains. Modern observations confirm that such dynamic thinning occurs over decadal timescales
(e.g., Felikson *et al*., 2017), but our data indicate that such processes may continue over centuries to millennia.
If the Grove Mountains are representative of the behavior of similar locations in interior East Antarctica, more of
the EAIS may have been thicker-than-present at the LGM and subsequent thinned than was previously thought.
Ice-sheet models may thus currently underestimate LGM ice volume and rates and magnitudes of deglacial ice
loss. Thicker-than-at-present LGM ice being limited to areas of East Antarctica within a few hundreds of
kilometers from the coastline would be consistent with reconstructions of MWP-1a that call for only a limited
input of meltwater from Antarctica (e.g., Yeung *et al*., 2019). Our work shows that EAIS thickening extended
further inland than indicated by [10]Be and [26]Al ages (e.g., Lilly *et al*., 2010), providing a modest additional ice
volume for MWP-1a, and that thinning started before and possibly occurred during the period of MWP-1a. While
we cannot accurately quantify how much EAIS volume was lost during this period, our data indicate that likely
less than half of the post-LGM ice loss occurred before or during MWP-1a in this region, consistent with studies
identifying Antarctica as likely being a minor contributor and the majority of the Antarctic contribution to have
been sourced from West Antarctica (e.g., Lin *et al*., 2021).
**5 Conclusions**
Our results using *in situ* [14]C provide new and improved constraints on past East Antarctic Ice Sheet thickness at a
site ~400 km inland from the present-day coast. These data show that the ice sheet at the Grove Mountains was
thicker, not thinner, at the LGM, but the summits of these nunataks were exposed. Ice-sheet thinning began here
~15 ka and continued through the Holocene, likely in response to changes near the grounding line that propagated
upstream. This work demonstrates that the LGM "hinge zone"—between thinner ice in the interior and thicker ice
at the coast relative to today—was further inland than was previously thought. The additional ice volume implied
by these findings therefore needs to be accounted for in numerical ice sheet and glacial isostatic adjustment
reconstructions of the last deglaciation.
**Code availability**
**Data availability**
All data described in the paper are included in Supplementary Table 1.
**Interactive computing environment**



**Sample availability**

**Video supplement**

**Author contributions**

CR processed samples for [14]C analysis, wrote the paper, and prepared all figures. CR, RJ, and AM conceived the project. All authors read and commented on the manuscript. BG provided code for exposure-age calculation and plotting. KL undertook fieldwork in the Grove Mountains and collected all the field observations and samples presented here.

**Competing interests**

The authors declare that they have no conflict of interest.

**Disclaimer**

**Acknowledgements**

This work was supported by Australian Research Council grants DE210101923, awarded to RSJ, and ARC Special Research Initiative 'Securing Antarctica's Environmental Future' (SR200100005). CR would also like to acknowledge support from the Monash Graduate and International Tuition Scholarships.

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
