# Peer review of "A thicker-than-present East Antarctic Ice Sheet plateau during"

_EGUsphere, 2024_

## Author Response (AR1)

**GB**

This is a valuable and fairly short and straightforward paper that should be published in approximately its present form. There are a lot of sites in interior Antarctica with Be-10/Al-26 data that are ambiguous as regards the LGM thickness, the only real way to fix this is with C-14 data, and that is what this paper does. The data here are useful and important.

There are a few things that could be improved, as follows.

1. The diagrams in Figures 1 and 4 that are supposed to show ice surface slopes would (and should) be greatly improved by drawing them with parabolic ice sheet profiles. As drawn with straight lines, it doesn't represent the concept of an LGM ice sheet with an extended grounding line but a thinner interior. For example:

[Figure]

At the very least, the 'coastline' and 'interior' labels appear to be in the wrong place on Figure 1c and need to be corrected. But currently the figures are barely acceptable -- they should be redrawn with more realistic profiles so that readers can understand what they are talking about.

==We agree with the comments, and revised the figure using more realistic parabolic profiles as suggested. See as follows:==

[Figure]

==We retain the straight lines in Fig. 4 to aid visibility but include the following in the caption of Fig. 4 to clarify this: "*Ice-sheet profiles have here been depicted as straight lines to aid visibility; the true ice-sheet profiles would curve as in Fig. 1(c)*".==

2. The discussion of the blank corrections needs some attention. First, the stated uncertainty of 3000 atoms is extremely small. Is this really the standard deviation of many process blanks? Second, the statement '..continually updated mean of all process blanks run at TUCNL since 2016...' somewhat conflicts with data in other publications, for example the appendix in this paper:

https://tc.copernicus.org/articles/17/1787/2023/

In particular look at Figure 1 in this:

https://tc.copernicus.org/preprints/tc-2022-172/tc-2022-172-AC2-supplement.pdf

The Tulane blanks shown here may have a mean near 58,000, but they have a standard deviation that is substantially (like 10x) larger than 3000. Furthermore, they are not normally distributed, so it would be inappropriate to use the standard error instead of the standard deviation, or to divide by sqrt(n). This issue doesn't really matter very much for the present paper (except regarding the fairly minor point about whether nonzero cosmogenic C-14 was actually observed in GR15), so the authors can do whatever they want here, but they need to explain what they did in more detail -- as written, the description of the blank correction is not acceptable.

We thank the reviewer for this observation. We changed the supplementary tables to reflect a blank of the mean value of the process blanks run concurrently with our samples and a blank uncertainty of the standard deviation. We also added Table S2 to the supplement, which presents the relevant blank data.  We updated the main text of the manuscript with the corrected sample concentrations and uncertainties.

3. The discussion of 'stratigraphic order' in various places (e.g., caption to Fig. 2 near line 182) doesn't really make any sense and needs some work. First, there isn't really any 'stratigraphy' here; what is actually being talked about is just the geometric constraint that lower-elevation samples can't be deglaciated unless higher-elevation samples have already done so. Second, depending on how you define the correct 'order', you could say that GR15 is out of order (if the order is defined by GR04, GR03, GR13, and GR01), or you could say that GR13, GR01, and GR12 are out of order (if the order is defined by GR04, GR03, and GR15), or you could pick some other things to be 'out of order' if you wanted. What the authors are trying to say here is pretty simple -- if the general deglaciation trend is defined by all the samples that are not GR15, then GR15 is out of order -- but the discussion here needs some attention.

We now clarify our reasoning for considering these samples outliers as: "*We consider samples GR12 and GR15 outliers, as the trend of decreasing elevation with decreasing age recorded by all of the erratic samples places these two samples out of order*".

4. In figure 3, the sample names should just be put on the figure contours, instead of complicatedly referenced in the caption. Also, labelling the upper right corner 'burial histories inconsistent with post-LGM exposure' is very confusing, because really these are just impossible burial histories: if burial began 15 ka, the duration of burial can't be any longer than 15 ka. It would be much clearer to just mark this 'impossible' or something of that nature, or (preferred) just make the figure border the correct shape to exclude this area entirely.. More seriously, though, it is not clear to me exactly what purpose this figure serves in the text. What point is intended to be made here? The reference to the figure in the text (line 219) suggests that it is supposed to indicate that

you can't get anywhere near saturation concentrations unless the burial took place a long time ago (lower left corner) or was very short (upper left corner). But nowhere is this really explained. In my opinion, this should be edited a bit to (i) move material from caption to text to make it clear what the point of this figure is; (ii) only include contours for the sample(s) that you are actually talking about, and (iii) generally clarify this discussion. Alternatively, this figure makes only a minor contribution to the discussion and can probably be removed entirely without significant loss of understanding.

Finally, I assume this figure assumes initial saturation at the time of burial -- that doesn't appear to be stated anywhere either.

We have updated this figure. The figure itself has been simplified, with the concentration contours removed, it focuses on only the difference between the saturated and non-saturated concentrations. The upper-right corner is labeled simply "Impossible burial histories", and we updated the figure caption to the following:

"*Figure 3: Burial-history contour plot. Contours show $^{14}C$ concentrations resulting from glacial histories assuming samples saturated with $^{14}C$ 50 ka, with one episode of burial under ice assumed to be sufficiently thick to reduce nuclide production in the sampled surface to negligible rates. The black-shaded part of the graph shows impossible histories (i.e., histories that require future burial). The grey-shaded part of the graph shows histories that would result in sample GR21 having a concentration below saturation for $^{14}C$. The unshaded portion of the graph shows the uncertainty window of a saturated sample at this latitude and elevation (7.3 x $10^5$ atoms $g^{-1}$; 72.9088 °S; 1,912 m a.s.l.). Only the lesser end of the saturation window is consistent with any significant degree of burial under enough ice to effectively stop production (~10 m); thus, only samples that were buried a long time ago or for a very short duration could show concentrations approaching saturation. Sample GR21 plots off the bottom-left corner of this figure; its mean 14C concentration (7.81 x $10^5$ atoms $g^{-1}$, see Table 1) is thus inconsistent with any episode of burial longer than 3 kyr in the last 30 ka, indicating that, if there was any significant duration of cover experienced by these samples, it occurred predominantly prior to the LGM. Permittable episodes of cover become shorter and occur earlier if samples are not assumed to be saturated at 50 ka.*".

Also a couple of minor comments:

Abstract, line 18. The 'relatively insensitive' here doesn't really make any sense. Be-10 and Al-26 are perfectly sensitive to short periods of exposure; the difference isn't the 'sensitivity', but the half-lives. It seems like what the authors are trying to say here is more like '...mostly come from surface exposure dating using cosmogenic nuclides with long half-lives, such as Be-10 and Al-26. These often record a cumulative exposure history extending over many glacial-interglacial cycles, rather than reflecting a single period of exposure after the most recent deglaciation....."

We have updated the sentence to say "*nuclides with long half-lives, which record ice-cover changes on timescales of tens of thousands of years and potentially multiple glacial cycles*".

Line 142. It would probably be helpful to mention that only the gas released at 1100 C is analysed; the 500 C step is for cleaning.

We agree and added this suggested text to the method.

Line 150. "run mass" doesn't really make sense here. I see that 'run' is supposed to be a passive voice verb, but it reads like a noun. Simply 'sample mass' would be much better.

We altered the phrasing as suggested.

Near line 204. The statement 'most post-LGM thinning is recorded during the Holocene' does not seem to be very meaningful, because most of the time since deglaciation is also in the Holocene (so this is kind of like saying that most of the numbers less than 10 are also less than 9). What is the point the authors are trying to make here? This could use some clarification.

We have added text clarifying that most of the thinning recorded since the LGM took place during the Holocene, after the earlier, late-Pleistocene stage of deglaciation, when the majority of Antarctic ice loss is recorded.

**AB-k**

This paper contributes to an important discussion about EAIS thickness during the LGM by adding new C-14 exposure ages, which generally circumvent the common issue of 10Be and 26Al inheritance in Antarctica. With these new data, the authors identify thicker-than-present LGM ice at a location previously thought to not have been covered by ice at that time. This finding has implications for EAIS volume during the LGM and the EAIS contribution to deglacial sea-level rise. The presence of samples with unsaturated C-14 samples also allows the authors to determine a post-LGM thinning history at this site, which couldn't be done with prior 10Be and 26Al measurements. Overall, I agree with the authors' treatment and interpretation of their data. I really enjoyed reading this manuscript – it is well written and nicely presented. There were a few points, however, that I think could be clarified with relatively minor revisions.

**General comments**

1. **Interpretation of saturated samples:** The authors spend some time with the question of whether the two saturated samples at/near the nunatak could have been covered by ice at some point during the LGM, which is certainly worthwhile as this determines whether they're able to put an upper bound on ice thickness during the LGM. They conclude that the answer is no, or if they were, it must have been for a short duration or by very thin ice (L217-219; L233-236) and use Figure 3 to support this conclusion. I really struggled, however, to digest Figure 3. There is a lot of information in this figure, so perhaps some of my questions below could be addressed with a slightly longer discussion in the main text and some updates to the figure.

   o The caption says the C-14 concentrations shown result from glacial histories with one episode of burial. I assume then followed by exposure so that the total history is equal to burial date? What are the starting conditions? I assumed saturation?
   The exposure history (what we assume is meant by "total history") would be equal to the difference between the burial date and duration, and the samples are assumed to start with no $^{14}$C (simulating lengthy burial during the LGM). We have edited this figure to assume saturation 50 ka. We edited the first line of the caption: "*Contours show $^{14}$C concentrations resulting from glacial histories assume samples saturated with $^{14}$C 50 ka, with one episode of burial under ice assumed to be sufficiently thick to reduce nuclide production in the sampled surface to negligible rates*". Assuming reset samples 50 ka does not substantially change this figure, due to the short half-life of $^{14}$C. We added the following to the caption to note this: "Permittable episodes of cover become shorter and occur earlier if samples are not assumed to be saturated at 50 ka.".

   o I was confused by the labeling of the black region as "Inconsistent with post-LGM exposure." Isn't the black region just above the 1:1 line for burial start and burial duration? Like, if burial started at 15 ka, and the burial duration was 20 kyr, the site would not only be buried today but also 5 kyr into the future? Since we know the sites aren't covered by ice today, they can't have the exposure histories that fall on that line, or anything in the black area because that requires future burial. In addition, the caption says that the black is scenarios that require burial after 15 ka, but I think there are scenarios that are not in the black area that also require burial after 15 ka? For example, 5 kyr of burial starting at 15 ka, or 10 kyr of burial starting at 20 ka.

We agree with these points, and changed the label to "Impossible histories" and adjusted the figure caption accordingly.

o What are the two different greys? Is this the region for burial histories producing unsaturated samples (stated in caption), and if so can the label be moved there? We updated this figure to focus more exclusively on the differences between the zones of saturated and unsaturated histories. The region with different gray shadings is thus one color and has been labeled as suggested.

o Is the entire white area the zone of saturation (as stated in the caption) or is it just the area to the left of the 7.3 x 10^5 atoms/g line (where the arrow and label point to)? If I'm reading Figure 2b correctly, it seems like saturation concentrations at 1921 m would span from the 7 x 10^5 atoms/g to the left side of the diagram? The white area is the zone of saturation. We have updated the figure to remove the concentration curves, which we felt distracted from the intended message of this figure.

o "Only the lesser end of the saturation window is consistent with any significant degree of burial under enough ice to effectively stop production (~10 m)" – I found this sentence confusing, probably in part because I was unsure what the bounds of the saturation window were (see bullet above). Is the "lesser" end just the lowest concentrations that are still considered saturated? If the entire white area produces saturated samples, then it looks to me that there's a lot of burial allowed. This sentence sort of makes it sound like the ice thickness needed to stop production was explored here, but I don't think it was? Maybe this just needs explanation in the first sentence of the caption – "…one episode of burial assuming that ice was thick enough (~10 m) that nuclide production in the sampled rock surfaces is negligible on these timescales." We agree and adjusted the figure caption accordingly (see our update to the first line of the caption, above). The figure itself does not explore the ice thickness needed to stop production.

o Is it possible to indicate where the saturated sample concentrations are on the diagram, rather than just referring to them being off the lower left corner of the diagram in the caption? Can the concentrations also be labeled with the sample id? While it is possible to include sample concentrations, in response to suggestions from other reviewers, we have elected to remove the sample concentrations entirely from the figure. Displaying mean saturation concentrations would require extending the figure substantially due to the asymptotic way that concentrations approach secular equilibrium – the closer to saturation a concentration gets, the longer ago burial of a given duration would have had to begin. Furthermore, we do not think that the specific sample concentrations contribute to our main conclusion of the figure, which is that any saturated sample would be consistent only with short burial or burial a long time ago.

o This figure actually opened a question for me about whether the unsaturated samples had inheritance, which I wasn't concerned about before seeing this figure. Considering an extreme example, the measured C-14 concentration in sample GR06 (highest unsaturated sample), could be achieved if burial started at 15 ka, with ~6 yr of burial and 8 kyr of exposure, meaning a true deglaciation age of 8 kyr. That scenario seems implausible, but is consistent with the data. On the other hand, as long as burial started before ~30–35ka, the apparent exposure age should be roughly equal to the true deglaciation age. I'm not sure if the authors were trying to make this point, but it came up for me in trying to

understand this figure.

This statement is true, but it is very difficult to confidently test whether the data reflect any inheritance. As we are unaware of any evidence of a glacial thickening ~15 ka elsewhere in this region, we have opted to present only naïve exposure ages for our unsaturated samples. Furthermore, glacial transport often results in erratics that have significantly scattered exposure ages relative to their elevations (as the $^{10}$Be and $^{26}$Al data do here). Our $^{14}$C data, however, show a mostly consistent trend of decreasing ages with elevation, with the two highest samples lying along a saturation curve.

Stepping back a bit, even if LGM ice-cover were compatible with the saturated C-14 concentrations, the authors' main conclusions still stand. The conclusion that ice was thicker at Nunatak 1921 during the LGM is still true and the unsaturated samples still record the thinning history after 15 ka. So maybe the level of detail in Figure 3, at least as presented, is just overcomplicating things a bit.

1. **MWP-1a discussion:** I actually find the sentences on L285-288 a more impactful way to end the Discussion, given the dataset and conclusions, than the discussion of MWP-1a, which I think could be shortened and simplified. There seems to be a tension between the fact that this chronology shows thinning during MWP-1a and its consistency with the EAIS as a whole being a minor MWP-1a contributor. I agree with the sentence on L290-292 that the work here suggests a modest additional ice volume for MWP-1a. I also agree that the chronology presented here suggests, although does not require, some thinning during MWP-1a (although "likely less than half of post-LGM ice loss" (L293-294) sounds like a lot of ice loss, maybe a nominal thickness loss (<20 m?) is a better reference here). However, this is one nunatak in one part of the EAIS, so I'm not sure it's necessary to extrapolate to the EAIS (L288-290) or Antarctica (L295) as a whole to the extent that's done here.

   We agree that a nominal value for the thickness of ice loss would avoid a potential misunderstanding. But if thinning during MWP-1a involved the ice thinning below the elevation of GR18 and then rethickening, our dataset (which only presents integrated exposure durations) would not be able to constrain the timing, magnitude, or duration of any such event. For this reason, we are reluctant to attempt to quantify this statement. We moved the MWP-1a discussion higher in our discussion section to emphasize the discussion of deglaciation and modelling, which we agree is more crucial to our findings.

**Minor Comments**

1. Figure 2 caption: I wasn't sure exactly what is meant by "error envelope" – is this determined by the typical measurement uncertainty, production rate uncertainty, or both (i.e., above this concentration there is no discernable change in the in the nuclide concentration beyond uncertainty)?

   The error envelope is defined by the instrument uncertainty. Its mean is the calculated secular equilibrium value at a given elevation and latitude assuming no erosion, and its upper and lower bounds are set by the instrumental uncertainty in repeated CRONUS-A measurements from the TUCNL. We added the following text to the caption: "the *vertical gray band to the right represents the saturation error envelope as calculated using the online exposure-age calculator formerly known as the CRONUS-Earth online exposure-age calculator using the CRONUS-A measurements listed in **Table S3**"* (lines 197-198).

2. L194–195: I might be careful about extrapolating to the covering of nunatak summits throughout the Grove Mountains as a whole, at least at this point in the

paper, because I'm guessing neither the elevation difference between each summit and the local ice surface, nor the change in LGM ice thickness, is uniform across the Grove Mountains. I also found the parenthetical statement here (neither lengthily nor deeply enough…" ) slightly confusing. Does this mean that if ice did override the summit, it wasn't thick enough to shield the sampled surfaces from the cosmic-ray flux?

Yes, that is our meaning there. We changed the wording to avoid any confusion.

3. L201–205: How were the percent thinning calculations made?
By assuming a linear thinning history (see lines 205-207 for further discussion).

4. L274-279: "Deglaciation thus possibly started and likely finished earlier downstream" - Are the Prince Charles Mountains actually downstream of the Grove Mountains (it doesn't look like it to me in Figure 1a)? I was also wondering if it is expected that the glacial history in the Grove Mountains is so different than in the Prince Charles Mountains, and if so, why? It looks like the White data are from Al-26 and Be-10, so is it possible they have some inheritance?

The northeastern Prince Charles are actually downstream of the Grove Mountains when considering that a modern flow path from the Grove Mountains to the nearest grounding line passes through them. We would expect thinning in the Prince Charles Mountains to predate that at the Grove Mountains due to the lesser distance from the grounding, meaning it would take more time for the thinning signal to propagate all the way to the Grove Mountains from a perturbation near the grounding line. It is possible that there is some inheritance in the White $et$ $al.$ (2011) dataset, but the relatively young (~20 ka) ages and the agreement between the two nuclides give us more confidence in the White $et$ $al.$ (2011) dataset than in the hundreds of ka ages ($^{10}$Be and $^{26}$Al) originally estimated from our samples (Lilly $et$ $al.$, 2010).

**Line edits**

1. L10-11: "380 km inland from the Antarctic coastline" – which sector of Antarctica?
This is in the Lambert Glacier–Amery Ice Shelf sector (drainage basin B-C, in Mouginot et al., 2017). We added the text explaining this (sans the parenthetical) to the abstract (lines 10-11).

2. L20–21: "above 1912 m asl"? Or, "from 1912 m asl to the nunatak summit at 1921 m asl"? Could this sentence also include an indication of how much thicker these findings require that the ice was during the LGM? Also, there is no mention anywhere in the abstract where Nunatak 1921 is – add reference to Grove mountains somewhere?
We changed this line to read "*Samples with $^{14}$C concentrations at a secular equilibrium between production and decay (saturation) at and above 1912 m a.s.l. indicate that the summit of a nunatak in the Grove Mountains was exposed during the LGM, requiring an ice surface ~70 m higher than at present*".

3. L59–59: Mention the half-lives of Be-10 and Al-26?
We added these half-lives to the text.

4. Figure 1 caption: move the sentence now on lines 84–85, which cites White et al. (2011) and Lilly et al. (2010), up to L68–69 to make it clear where this placement of the potential hinge zone comes from?
We moved this sentence to lines 73-77.

5. Figure 1c: "interior" and "coastline" are switched.
We corrected these labels.

6. Line 87–88: "testing previously measured samples at a key site in the ice sheet interior" – maybe just state what you are testing and what the key site is?
We rephrased the sentence to "*…by measuring in-situ $^{14}$C in bedrock and erratic samples previously measured for $^{10}$Be and $^{26}$Al from the Grove Mountains, a key site in the ice-sheet interior.*".

7. Section 1.1: Make it clear that Nunatak 1921 is named for the altitude of its peak and also state the ice surface elevation at this site specifically? I think it's mentioned later but it would be helpful to have it here.
We added this information in parentheses to the text.

8. Table 2 caption (L160): 10^5 is missing when stating blank value.
We added this value to the table caption.

9. L167: 10Be and 26Al exposure ages, not concentrations.
We made the correction.

10. L182: GR12, not GR21, is out of order?
We corrected the naming.

11. L202: Add timing of MWP-1a since this is the first mention? Also, should the reference be to figure 2d, not 2b?
Timing added and figure-reference corrected.

12. Line 210: Maybe specify at Nunatak 1921, instead of in the Grove Mountains generally? As consistent with a few comments above, this could be done more often throughout the paper.
We changed the text accordingly to specify the nunatak.

13. Line 212: "contrary to previous ice-thickness data" – this isn't really true, it's contrary to previous interpretations of 26Al and 10Be data.
We adopted the suggested phrasing.

14. L216: "indicate that ice cover occurred at this site [up to x m above the present ice surface]?"
We added this information to the text.

15. L219: "re-saturated during the Holocene" – or during the deglacial / late glacial and Holocene, if it must have been uncovered before GR06?
We changed the text from "during the Holocene" to "following re-exposure".

16. Figure 3: 14.9 kyr stated for GR06 in caption, but table and text say 14.6 kyr. Also, the caption says GR21 = 7 x 10^5 atoms/g, which I don't think is right?
We revised this figure and removed references to individual sample concentrations from its caption.

17. L247: Rather than the coast being representative of the interior, could this be simplified to "the zone of thicker-than-present LGM ice extends further inland than previously thought"?
We simplified the wording based on this suggestion: "*Our new chronology indicates that the zone of thicker-than-present LGM ice extended further inland than was previously thought*".

**Citation**: https://doi.org/10.5194/egusphere-2024-2674-RC2

**AR#3**

GENERAL COMMENTS

This manuscript presents in situ 14C measurements from an altitude transect of bedrock and erratic samples from the Grove Mountains in East Antarctica that had previously been measured for in situ 10Be and 26Al by Lilly et al. (2010). That earlier study found large inherited inventories of the longer-lived nuclides. In situ 14C provides a means of seeing through that inherited signal to try to discern evidence of deglaciation since the Last Glacial Maximum (LGM) at ca. 21 ka. The results could indicate whether ice sheet models that predict coastal thickening and inland thinning of the East Antarctic Ice Sheet at the LGM are consistent with empirical evidence. This study constrains ice thickness history at a location significantly inland from the East Antarctic coast, but in my view requires significant additional data and discussion in two main areas before publication would be appropriate.

Most importantly, in my view the manuscript suffers from sloppiness and inconsistencies in the descriptions of the procedures used, as well as in presentation of the data and assumptions made in their interpretation. Clarity and transparency in those areas is critical to allow comparison with other work. It's clear that procedures have changed since the original Goehring et al. (2019) Tulane laboratory study; however, I could not find supporting data for those changes in the literature (see Specific Comments below). Much of the interpretation (e.g., production rates, saturation concentrations, etc.) is based on measurements from the Goehring et al. (2019) study. Without detailing potential effects of the subsequently modified procedures on measured 14C concentrations from more recent replicate analyses of CRONUS-A or other intercomparison materials (or otherwise documenting that there are none), this leads me to question the implied robustness of the results presented here. It's not enough just to state that there are no effects from the procedural changes – that conclusion should be documented. Neglecting to discuss potential additional uncertainties (if any) arising from these procedural modifications and calculator dependencies (see Specific Comments) gives the impression of in situ 14C ages that are more robust than they actually are in my view – particularly for pre-Holocene ages. I'm happy to be proven wrong about this impression but the effects of procedural changes on measurements and interpretations should be well documented.

We apologise for some inconsistencies in the description of our procedure, and have added more detail and made some corrections to the text. Additional CRONUS-A measurements were performed at the TUCNL, and we incorporate these new data into our production-rate calculations and recalibrate all the $^{14}$C ages of our samples. Using the additional CRONUS-A measurements together with the online UW calculator does not significantly alter the calculated ages. Importantly, these adjustments to ages do not change the narrative and conclusions of this paper. We have added an additional table to the supplement documenting the measurements used to determine the production rate for our samples. We have added text acknowledging the CRONUS-A production rate and calculation method used to the caption of **Table 2**.

Second, I think that the study would have benefited from at least one bedrock-erratic pair being analyzed from the original paper by Lilly et al. (2010), with more justification for sample selection (if there was no sample material left for paired analyses, fine, but that should be stated explicitly). Only three bedrock samples were analyzed, and not in conjunction with their respective erratics from the earlier Lilly et al. (2010) study. Analyses in that original study were split between 16 bedrock and 10 erratic samples, with several bedrock-erratic pairs – most contained significant inherited 10Be/26Al signals relative to the 14C results here. Given the analytical focus here on erratics and the fact that the Grove Mountains comprise an isolated small group of nunataks with varying compositions (GeoMap - https://data.gns.cri.nz/ata_geomap/index.html) subject to W-NW ice flow toward the Lambert Glacier and Amery Ice Shelf (e.g., Mouginot et al., 2019, Geophysical Research Letters, 46(16), 9710–9718), a more detailed discussion of the site setting in terms of ice flow and broadly up-gradient lithologies would be useful. In particular, a discussion of rock types comprising the local nunatak group shown in Fig 1b would be very helpful to place the erratic lithologies in context to assess the possibility of exposure prior to deposition – particularly for those nunataks broadly up-gradient from the sampled nunatak (see Specific Comments). In my opinion, the authors need to demonstrate to the extent possible that the erratics are indeed truly erratic, and not locally derived lithologies that may have been exposed supraglacially during transport or recently on hillslopes prior to deposition on bedrock at the sampled locations, given the clear long-term inherited signals. For example, were there striations or other evidence of subglacial transport associated with the cobbles? If there is none, that's fine, but then that should be stated and clarified that the authors are then simply assuming that they are true erratics. Descriptions of each sample (with photos if possible) in the supplement would be useful information in this context. Analysis of bedrock-erratic pairs from the same locations would have gone a long way to support the authors' arguments. Given the remoteness of the study area and 20-20 hindsight from a decade and a half ago, I'm willing to cut the authors some slack on these points, but I believe that at least some non-trivial discussion of ice flow and local bedrock lithology from the local nunataks and the potential for prior erratic exposure over the late Pleistocene/Holocene is warranted.

This is a good point, and we have made several edits in light of it. Additional data, especially bedrock-erratic pairs, would be very helpful here. The ten samples presented in this study were, however, the only samples that had sufficient material remaining to analyze. The erratic samples measured in this study had already been partially processed prior to the beginning of this study, and had already been crushed, preventing us from identifying any signs of subglacial transport (if there were any). We added language to the manuscript acknowledging the gaps and uncertainties inherent in this dataset. We altered text that implied a definite erratic origin of these samples, instead noting that the samples likely did not originate from the nunatak on which they were collected. If the samples were derived from a nunatak up-glacier of Nunatak 1921 in the Grove Mountains, they could have traveled a limited distance before being deposited. In this case, any inherited $^{14}$C in our samples would imply that the ice-sheet thickening (not revealed by the $^{10}$Be and $^{26}$Al in our samples) was larger or more recent than assuming that the $^{14}$C concentrations of the samples were reset prior to their most recent exposure would. In the absence of evidence which could confirm their origin, we have made the more conservative assumption regarding the magnitude of ice-sheet thinning since the LGM.

Once these significant issues (and Specific Comments below) are addressed, I look forward to reviewing this paper again.

SPECIFIC COMMENTS

Line 1:     Thicker and thinner are relative to something. Suggest changing the title to 'A thicker than present East Antarctic Ice Sheet plateau during the Last Glacial Maximum'

We agree and changed the title to "A thicker-than-present East Antarctic Ice Sheet plateau during the Last Glacial Maximum".

Line 9:     in situ is not hyphenated as it is Latin – should just be italicized throughout

We acknowledge this point but prefer to keep "*in-situ*" hyphenated here because it is modernized Latin, forming a compound adjective.

Line 14:     On the other hand, 'ice sheet' SHOULD be hyphenated, as it modifies the word 'interior.' The rule is that compound words that act as a noun such as 'ice sheet' should not be hyphenated, but should be hyphenated when they modify another noun (serving as an adjective), like 'ice-sheet interior', 'ice-sheet history', 'ice-sheet model', etc. Be consistent throughout.

We agree and changed all instances that do not form part of a proper-noun phrase.

Line 20:     'Saturated' is colloquial - better to avoid colloquial phrasing without first defining. Better to say "Samples with 14C concentrations at a secular equilibrium between production and decay (saturation)..."

We made the suggested changes to first define 'saturation'.

Line 43:     Should use Oxford commas throughout in lists – 'radar, ice-sheet, and geological…'. Suggest replacing 'sparse', with 'rare' since 'sparsity' is used later in the sentence

We inserted the Oxford comma. While changing "sparse" here to avoid duplication is a good suggestion, we are attempting specifically to draw attention to how geographically dispersed this data are, not the lack of data itself, so we feel that "rare" risks causing some readers to misinterpret our meaning here.  Other phrases that communicate the same thing as "sparse" lack the clarity and word economy of "sparse" here, so we prefer to retain this phrasing.

Line 44:     Active voice is almost always better in writing: ''often disagree' instead of 'are often in disagreement…'

While active voice *is* almost-always better in writing, in this case, the data (unlike, e.g., the researchers who collected them) are themselves inanimate records incapable of actively disagreeing with one another.  We thus consider passive voice more appropriate here.

Line 51: Replace 'indicate with 'suggest'. Less definitive since an interpolated value is referenced

We made these changes.

Line 58: Replace 'Yet' with 'However,'.

We changed the line to "*Existing cosmogenic-nuclide data from regions of cold-based non-erosive ice, however,…*".

Line 60: Replace 'nuclides' with 'nuclide inventories'. More precise wording

We have made the suggested change.

Line 62: Reference should just be to Goehring et al. (2019) – no Balco. If there are multiple Goehring et al. (2019) citations for different papers, then specify 2019a, 2019b, etc.

We added numeric codes to the citations to differentiate them.

Line 65: Fig 1c appears to have the Interior and Coastline labels reversed, based on the caption

We made this correction.

Line 73: Shading looks more pink than red

The text describing the figure color is now "Pink".

Line 77: Fig 1b: Would be useful to indicate the general direction of ice flow on this image - what nunataks are upstream of the sampled nunatak, if any

We agree and added an arrow to the figure, with text added to the caption: "Ice at this site flows slowly (blue arrow; Rignot et al., 2011) northwest, towards the Amery Ice Sheet, though flow speeds are low and directions are strongly influenced by topography in the vicinity of nunataks (Lilly, 2008).".

Line 89: Instead of 'be saturated with' it would be more accurate and succinct to say here 'have concentrations of in situ 14C at secular equilibrium between production and decay (saturation) - a state that requires at ca. 5 half-lives of continuous exposure (Dunai, 2010). Need to make clear that it's the concentrations that indicate secular equilibrium.

We agree and made the suggested change.

Line 90: Delete 'exposed'

We deleted this word.

Line 91:     Replace 'covered at some time since the LGM' with 'likely covered for some duration post-LGM'. More precise wording.

We agree and made this change.

Line 107:     This is the situation I referenced in my General Comments. Do the rock types of the cobbles occur in the Grove Mountains (or is bedrock all just orthogneiss throughout)? The GeoMAP site, (https://data.gns.cri.nz/ata_geomap/index.html) indicates felsic plutonic rock types are actually quite common in the vicinity of the sampled nunatak. While that map is at quite a large scale, this suggests that at least some of the cobbles could well be locally derived erratics and thus have the potential for some subaerial exposure either on the land surface (e.g., rockfall and downslope transport) and/or on the ice surface (e.g., rockfalls onto the ice) before being deposited at the sampled locations. I'm not convinced by the single declaration that they are not locally derived, without any other discussion. Are the erratics striated or otherwise have evidence of subglacial transport? The rock types in Table 1 indicate orthogneiss for the bedrock, but all the erratics could totally be associated with felsic plutonics in the vicinity, lacking any more detailed description of the rocks. If the quartzites are sedimentary, state that as that is evidence of a true erratic. Metamorphic quartzites could potentially be associated with the gneiss. My point here is that uncertainty is fine but one needs to be up front about it. Statements of certainty when in fact significant uncertainty exists is a common theme I find in this manuscript.

We agree that some of this wording could be clarified, and we have changed "locally derived" to "derived from Nunatak 1921".

Regarding the details of the rock type, we are unfortunately limited by what information is available for the samples and study site. Orthogneiss is the only bedrock type known to us for these nunataks (e.g., Lilly, 2008; Lilly *et al.*, 2010). Unfortunately, all of the erratic samples had already been crushed and sieved by the time we accessed them for this study, preventing us from identifying surface features that could support subglacial transport. Sample notes do not mention striae, etc. on the erratic samples, but we cannot rule out such erosional characteristics. We are unable to make any association between the quartzites and the local orthogneiss.  While the felsic metamorphic samples could indeed be related to nearby (e.g., charnockite) exposures, we lack sufficient evidence to tie our samples to any known outcrop, as well as evidence indicating that they were sourced from exposed rock rather than a buried occurrence of similar rock either nearby or further afield in the East Antarctic interior. While ice flow through the Grove Mountains is not in simple, the general flow direction to the north and west narrows the range of nearby nunataks which could act as source regions for these samples.  The Grove Mountains do not cover a large area; a locally derived erratic could have travelled atop a glacier for <10 km at most before being deposited on Nunatak 1921, but the slow ice velocities and young ages in our lower-elevation samples mean that this could still contribute a portion of the measured nuclide inventory.  Assuming any subaerial exposure during transport would imply additional ice-sheet thinning since the LGM for which we would lack evidence – as we can only claim with certainty that the samples are not from Nunatak 1921, assuming subglacial transport is the more conservative assumption. We have added additional discussion to lines 115-117 that provides this context,

acknowledging that we cannot be certain about the provenance of the erratic samples. The main conclusion of our study, that the EAIS at this location was thicker than it is at present at the LGM, is unchanged either way – ice would have to have been thick enough to deposit the samples at their current location regardless of whether the samples were transported supra- or subglacially.

Line 114:   Indicate the half-life for each nuclide considered so that the reader can assess the duration required for saturation for each.

We agree. The half-lives of these nuclides have been added earlier in the manuscript.

Line 123:   Looks as though from Fig 2 that 4 of the erratics were part of bedrock-erratic pairs from the original paper? Why were the corresponding bedrock or erratic samples not analyzed here - at least one or two of them? In those cases, the higher altitude erratics generally indicated longer apparent exposures than the corresponding bedrock. It would have been very useful to have that perspective for this dataset as well. State the reason(s) both members of the pairs were not analyzed to clarify for the reader. See comment on Line 107.

While bedrock-erratic pairs would definitely have further strengthened our understanding of this site, unfortunately, only material from those ten samples were available for this study. We acknowledge this in the text.

Line 128:   Is the lithology actually 'Unknown' or just not recorded and no unprocessed sample remaining? If the latter it's better to just say that. If the latter is not the case then it should be possible to ascertain some sort of rock type for the sample.

Unfortunately, all the erratic samples measured in this study arrived in our possession already crushed. The crushed sample contained quartz, feldspar, and mafic minerals, but no lithology could be positively identified. The "Lithology" has been changed to "Unrecorded", and we added a refer to this table noting the sample mineralogies.

Line 141:   'Li-flux-containing' is awkward wording. Clarify to indicate that the flux had been previously fused and degassed of contaminants and cooled prior to loading.

Crucibles are typically round – these should be referred to as Pt combustion boats.

We changed the wording as suggested.

Line 142:   Goehring et al. (2019, Nuclear Instruments and Methods in Physics Research B, 455, 284–292) indicate combustion for 1 hr at 500 C and extraction for 3 hr at 1100 C. However, Nichols and Goehring (2019, Geochronology, 1(1), 43–52) subsequently indicated a procedure of 30 min at 500 C followed by 3 hr at 1100 C – this change in combustion procedure that potentially has implications for the extracted 14C results (e.g., Lifton et al., 2023, Geochronology, 5, 361–375) was presented without supporting experiments. Analyses presented in Balco et al. (2023, The Cryosphere, 17(4), 1787–1801), coeval with the samples from this study, indicate combustions of 30 min at 500 C with extractions of 2 hr at 1100 C,

again without experimental data supporting the procedural change. It's thus unclear to me what procedure was actually followed here in these analyses, as the authors cite only Goehring et al. (2019). Given that these procedural changes have the potential to affect the measured 14C concentrations significantly in either direction, I think it's crucial to clarify what procedures were actually used in the analyses here, and to the extent that they deviate from the procedures in Goehring et al. (2019), to document that the modified procedures had no significant effect on the resulting measurements.

The procedure as laid out in the manuscript specifies 30 min of heating at 500 °C, followed by 3 h at 1,100 °C, consistent with that described in Nichols and Goehring (2019). We therefore also cite Nichols and Goehring (2019) in the methods and altered the associated wording.

Line 143:   Specify what is meant by 'hot', and describe the quartz for the reader - single crystal, gravel, sand, etc. - provide citation

We specify that the quartz used is chips at 850 °C, as per Goehring *et al.*, 2019, which we cite.

Line 144:   Specify to what equivalent mass of C the sample is typically diluted to.

The mass of C in the sample is diluted to ~110 μg, which we added to the text.

Line 146:   NOSAMS measures the 14/12 isotope ratios, not the concentrations. Concentrations are derived from those ratios - reference the data tables. Clarify.

We acknowledge that this wording is misleading and clarified the description and supplementary table.

Line 148:   As noted in Balco et al. (2023), and in Greg Balco's review comments, this is not necessarily representative of the system blanks at the time of extraction for this study's samples. The blank data included in Balco et al. (2023) show wide temporal variability - the 3110-atom uncertainty is not representative – the standard deviation of the data is about 10x that, as indicated by Balco's comments. Also indicated by Balco, if this is standard error, that is not appropriate for a non-Gaussian distribution such as that of the blanks. Comparing the sample numbers (TUCNL) from this manuscript's supplement with those from the Balco et al. (2023) supplement, it appears that these data were coeval with some of the Balco et al. data. In my opinion the most defensible approach is to utilize only the blanks from the time of these analyses. See my comments on the Supplemental data for more details.

We agree that the blank correction should represent the time of these analyses.. Follwing this suggestion, and on the advice of other reviewers, this sentence has been changed to "*…of 58,000 ± 12,600 atoms was subtracted from the total measured atoms from each sample; this value represents the standard deviation of process blanks run at the TUCNL (Goehring et al., 2019) over the timespan within which samples for this study were measured (July 10, 2021- August 27, 2021).*"  We have updated all sample concentrations and uncertainties in the text and supplement accordingly, but note that the changed values do not impact the conclusions of the paper.

Line 155:    Again, see the Balco et al. (2023) 14C supplement. The measurements of CRONUS-A since those included in Goehring et al. (2019) scatter quite a bit more than what is quoted here and show a significant uptick in concentration in the most recent two values in that paper (but which are still well before these analyses). Given that the Goehring et al. (2010) CRONUS-A measurements are used as the basis for the default 14C production rate in the University of Washington v3 online calculator (UWv3 - Balco et al., 2008), and particularly in light of the last two higher concentrations in Balco et al. (2023), the authors should present any subsequent CRONUS-A measurements spanning the time that the Grove mountains samples were run to demonstrate that the production rates assumed are appropriate. If the high concentrations of more recent samples are more representative of results when these samples were run, then a production rate consistent with those should be used, with associated changes in ages and predicted saturation concentrations.

The authors need to be fully transparent about their data and underlying procedures/measurements since the production rates used (and subsequent exposure age implications) depend on the significantly (15-20%) lower measured concentrations of CRONUS-A in Goehring et al. (2019) as compared to most other labs (e.g., Lifton et al, 2023).

Also, the authors should post the code being used to calculate the ages - running the concentrations and site parameters through the UWv3 calculator gives quite different ages for the older ages, and larger uncertainties as well than what are presented here. Any conclusions based on ages thus need to be approached very cautiously as pre-Holocene ages appear less robust than the late Holocene ages.

We incorporate the additional CRONUS-A measurements performed at the TUCNL and recalculate our ages using the code from the online UWv3 calculator, using the additional production-rate data, and clarify this in the methods section (line 158). As mentioned above, adding a small number of additional CRONUS-A measurements insignificantly alters the calculated ages. The only finding that stood to change based on the ages of the pre-Holocene unsaturated samples is in whether or not samples GR04 and GR18 conclusively postdate MWP-1a – our discussion of this point has been modified slightly to reflect the revised uncertainty windows of these samples. We added an additional table to the supplement documenting the CRONUS-A measurements used to determine the production rate for our samples.

Line 156:    It is not clear where this claimed 6% uncertainty comes from. The CRONUS-A concentrations in Balco et al. (2023) have a standard deviation of about 8-9% - if one includes the last two from that paper which are much higher than most of the others, it's over 10%. Also, the 6% uncertainty in 14C concentrations is almost certainly concentration-dependent - see Balco et al. (2023) for example with low-concentration samples' % reproducibility - this should be stated as such. CRONUS-A is a high concentration sample - some of these concentrations are high as well but many are much lower, so that uncertainty is likely a minimum value in my opinion. Clarify.

The 6% uncertainty comes from repeated measurements of CRONUS-A material on the TU-CEGS ("at the TUCNL") (see lines 155-157). These show reproducibilities of ~6%; therefore,

we use this value as the most appropriate error window for our measurements. We adopt this method following its use in other studies published on samples from this lab – see, for example, Nichols *et al.* (2019).

Line 160:    A blank of $0.58 \pm 0.31$ atoms is inconsistent with the text – also see above comments. Make sure the units and values are correct and clearly stated, and reference supplemental data tables for complete information.

We agree and corrected the blank uncertainty to 0.13 in the text and supplementary table referenced in caption.

Line 161:    What level of uncertainty is being cited here and throughout? 1 sigma, 2 sigma? An early statement as to what level all uncertainty values represent will take care of them for the whole paper, unless otherwise noted.

We now specify one sigma in the text.

Line 165:    $0.02 \pm 0.01$ ka? 20 years? I don't believe it. Running the data through the UW v3 calculator yields ca. 200 yr. Proofread all numbers in tables and text.

We apologise for this error and have corrected all floating-point typos.

Line 169:    Is there field evidence to support the claim of sediment or boulder cover that moved recently? Pretty speculative. Also, is there evidence of drifting snow in the sample vicinity currently? State clearly that this is speculative. Again, it's fine to speculate or have uncertainty just be up front about it clearly. Particularly when you already said there's no evidence of past snow or sediment cover.

We can unfortunately only be speculative as we have no field notes from the researchers who collected these samples. We acknowledge that this is speculative in the text.

Line 174:    Again, what significance (or sigma uncertainty) level is indicated by the error bars? 1 sigma? 2 sigma?

Our uncertainties are presented at one sigma, as noted above. The levels of the $^{10}$Be and $^{26}$Al uncertainties from the original studies are not reported, so we assume they are 1 sigma uncertainties as is typical when reporting cosmogenic nuclide concentrations and exposure ages.

'Approximately' is a more precise term than 'roughly'

We agree and have changed the wording.

Line 176:    As implemented in this figure, the highlighting makes the numbers harder to read. It would be better to make the sample numbers from this study in a different font, color, bold or italic, etc., to set them apart more clearly. You could also make the photo larger and

more legible if you move the legend into panel d, for example, or otherwise rearrange the legend.

The samples measured in this study are marked with outlined yellow boxes, in larger, bolded font; the photo has been enlarged and legend shortened and widened to accommodate the change.

Line 178:    Again, what significance (or sigma uncertainty) level is indicated by the error envelope? 1 sigma? 2 sigma? Based on what production rate? Specify

The error envelope here represents the uncertainty on repeated CRONUS-A measurements at the TUCNL (5.6%, rounded to 6% for reporting in text). We specify in the text that this value is as calculated using the online UW calculator.

Line 194:    'Lengthily' is not a word. Suggest rephrasing to something like '...but not thick enough to override the summits for a long enough duration to allow the 14C to decay below saturation'

While we disagee that "lengthily" is not a word, we acknowledge that it may nevertheless cause confusion for some readers.  We have changed the wording to use more-common language.

Line 201:    14.9 ± 1.0 is inconsistent with what is listed in Table 2. And as noted previously, pre-Holocene ages appear to be calculator-dependent, so any correlations with well-defined events such as MWP-1a should be appropriately couched in language emphasizing the uncertainties.

We agree and have changed the wording to more-clearly emphasize the uncertainties in the dataset.

Line 214:    Delete 'do not show saturation' and instead reference that many of the samples from the original Lilly et al. (2010) paper show evidence of complex exposure over long time frames (and then state the range of minimum exposure durations consistent with each sample)

As suggested, we refer to the original samples and deleted that saturation wording. The oldest sample (GR15) has a minimum $^{10}Be/^{26}Al$ exposure duration greater than the maximum possible for, for example, GR01; thus, no range of minimum exposure durations is consistent with *each* sample.  We thus think it most appropriate to only note the evidence long, complex exposure histories for our samples and then reference Table S5, which lists the $^{10}Be$ and $^{26}Al$ ages of these sample, rather than listing them in the main text.

Line 216:    Suggest 'Our 14C data indicate that this site was covered by ice > ca. 10 m thick for long enough to allow in situ 14C concentrations in the samples to decay.' To what level, though? Measurement background – what is that for this case? Specify. It depends on how

long the shielding lasted, and starting from what concentration? What are you assuming here - justify that.

We added details specifying the assumptions that samples were $^{14}$C-saturated prior to cover and that cover was deep and long enough for concentrations to decay to near-background levels; although the degree and magnitude of cover will have varied by elevation (see Fig. 3).

Line 218: What do you mean specifically by 'covered briefly'? Put a value on this - if the summit was covered by over 10 m of ice, how long could it have been covered during the LGM or subsequently and still yield saturated concentrations today?

We now specify the value in the text (≲3 kyr in the last 30 kyr, as per Fig. 3). However, note that, the further in the past that cover occurred, the longer it could have lasted while still allowing summit samples to exhibit saturation today.

Line 219: Suggest '… nunataks were progressively re-exposed through the late Holocene.' More succinct.

We agree and have made this change.

Line 223: What is the starting point for these calculations - do you assume all samples were saturated before burial? What effect does starting from a non-saturated concentration have on the predictions here? Describe for the reader.

On the advice of other reviewers, this figure has been revised. The caption specifies that it assumes saturation 50 ka, and that assuming initial exposure only makes permittable episodes of cover shorter and earlier. The minimum burial age resulting in saturation decreases by up to 5 kyr in some regions of the graph space, but more than 3 kyr of burial in the last 30 ka still results in unsaturated summit samples.

Line 230: Again, the 14.9 ka age is calculator-dependent. Provide specifics and discuss effects of different calculated ages for that sample (a different production rate and calculation scheme would also likely affect this plot overall)

The purpose of this figure is to demonstrate whether it is possible for an exposure history to plot in this region of the graph space, and the absolute age is here irrelevant; a sample buried 30 ka can have been buried for at most 30 consecutive kyr. We have therefore removed the age from the text, and the figure and its caption are altered accordingly.

There appear to be three or four slightly different shades of gray on the graph, 'gray-shaded' could be more specific. Might be more obvious for the reader if colors were also used instead of just grayscale.

On the advice of other reviewers, the colors in this figure are updated, consisting of only one shade of grey for clarity.

Line 231:   '…being unsaturated with 14C' is awkwardly worded. Suggest 'having a concentration below saturation for 14C'

We agree and changed the wording as suggested.

'The unshaded portion of the graph…': As above, define where this is - it's not clear from the figure in the PDF - does everything have some degree of shading except for concentrations >7.3e5? Suggest modifying the shading (colors or something more obvious to the reader than really light grays - bigger steps between grayscale values would help). Maybe hatchures instead of black in forbidden region. Describe what is meant by 'uncertainty window' - it should reflect Fig 2b, but does not appear to with this shading scheme. Are you indicating any concentration > 6e5, per Fig 2? Make sure all figures and discussion/descriptions are internally consistent.

Yes, the line between the white and grey regions of the graph now represents the $7.81 \times 10^5$ atom g$^{-1}$ contour – we state this in the Fig.3 caption.

Line 238/39:   Be skeptical of all 10/26 ages in cold-based regimes such as many places in Antarctica. It obviously would be good to have 14C from these datasets if possible now as well to confirm there's no significant inherited inventory (not 10s-100s of ka, obviously, but perhaps a few ka worth). Be up front as to the potential pitfalls of relying on long-lived nuclide chronologies in these environments - it may be that the ages are fine once all recalculated using the same methods/assumptions (e.g., UWv3 calculator, LSDn), but to me there is always going to be some uncertainty in 10/26 ages for LGM and younger time frames in Antarctica, without 14C confirmation. Future 14C work, yes, but point out the potential for even low levels of Be inheritance.

This is a very good point. We added text noting the potential for inheritance in these regimes: "*Direct constraints from cosmogenic $^{10}$Be and $^{26}$Al show evidence of the ice being thicker near the Antarctic coast (e.g., Mackintosh et al., 2007; White et al., 2011a), but exposure ages derived from the same nuclides from interior sites such as the Grove Mountains pre-date the LGM*" (lines 264-266). We also note the usefulness of $^{14}$C in checking for it (see later comment).

The authors should also provide code for calculations through GitHub/Zenodo or similar, as noted previously, so that interested readers can work to reproduce all calculations and results here.

Line 243/44:   See above comment - rephrase this paragraph to clarify the possibility of even low levels of inheritance in longer lived nuclides.

We added a note here that acknowledges this point: "*The potential for low levels of $^{26}$Al and $^{10}$Be inheritance in cold, arid regions highlights the usefulness of $^{14}$C as a tool for improving ice histories derived from long-lived nuclides.*".

Line 248:   Prefer 'geomorphic' to 'geomorphological'

We made this change.

Line 257:     Fig 4: 'Present-day ice surface' should be located along the actual line for clarity - above the LGM surfaces on the right side would work. Alternatively, have a legend on the figure identifying each dashed line separately from the lines themselves. If the present-day ice surface continues below the previous LGM surfaces, then it should continue into the gray area at the bottom of the figure.

We moved the line labels to right of the figure and extended the line of the present-day ice-surface into the grey area at figure bottom.

Line 261:     Is the ice surface slope upstream of the hinge zone constrained by the Prince Charles Mountains data as you have drawn this, or could it just be subparallel to the previous LGM surface upstream of the hinge? Clarify in the caption and discuss further in the text.

We adjusted the ice-surface slope upstream of the hinge zone as suggested, as this would indeed be more realistic.  We also mention in the caption that neither distances nor angles in this figure are to scale.

Line 268:     Reword - this is a confusing sentence. Suggest '...EAIS being thicker than previously suggested at the LGM is that...' Any leads and lags should be evaluated with 14C in coast and interior locations to reduce the possibility of minor inheritance in longer-lived nuclides. At least you should qualify any discussion with that possibility so the reader is clear on that.

We changed the wording as suggested. We also note in the paragraph that we compare $^{14}$C data from the Grove Mountains to sites that lack $^{14}$C data.

Line 271:     Clarify which White et al. (2011) you're citing in each case - should be 2011a and 2011b to differentiate. The authors cite two.

We clarify these citations.

Line 274:     As noted earlier - be careful tying this 15 ka and other 14C ages to other events as especially the older ones are calculator dependent. Make sure all previously published ages in this paper (14C, 10Be, 26Al, etc.) are re-calculated using the same underlying assumptions and algorithms, and make sure to state that that is what has been done. The UW v3 calculator and ICE-D Antarctica is quite useful for that. And specify which production rate datasets you're using for each nuclide.

We agree and recalculate the $^{10}$Be ages presented for the sites mentioned in the text. We also add a note stating how these were recalculated (using the ICE-D calculation framework built on UW v3 [Balco et al., 2020]).

Line 276:     Again, my take on these 10Be ages is that they should be viewed with caution as they can easily skew a bit old - hence the importance of 14C. Any comparison between 14C

results and 10/26 results from other sites should be appropriately qualified in the discussion - that there is the potential for 10Be/26Al ages (even post LGM deglaciation ages) to skew older due to an inherited component. I don't think you can get away from that possibility. And especially since the 14C ages and uncertainties are calculator-dependent to some extent, the authors need to dial back strong correlations.

We agree that $^{10}$Be ages should be viewed with caution and tried to reflect that in our wording. For example, the phrasing "…elevation was reached by…" refers to conservative estimates, allowing for the possibility of inheritance increasing the apparent exposure age. We modified the wording regarding our discussion of the lag time between deglaciation of the Prince Charles Mountains and Grove Mountains to better acknowledge the limitations of the $^{10}$Be data. However, we think that future $^{14}$C deglaciation ages in the Prince Charles Mountains are unlikely to overturn our conclusion that deglaciation began earliest near the Lambert Glacier grounding zone and propagated up-glacier. We have added a sentence to this paragraph explaining that the longer-lived-nuclide ages may decrease as additional data becomes available.

Line 277:    Clarify which White et al. (2011) citation – there are 2 such papers cited.

We now specify more clearly.

Line 286:    Replace 'thinned' with 'thinned more'

We replaced "subsequent thinned" with "subsequently thinned more".

Line 288:    Replace 'thicker-than-at-present' with 'thicker-than-present'

The beginning of this sentence has been changed to "Ice in East Antarctica being thicker at the LGM than at present only within…".

Replace 'hundreds of' with 'hundred'

We have changed this wording as suggested.

Overall this sentence is the sort of thing I'm talking about in my earlier comments - yes using the calculator you are employing gives something close to MWP-1A, but UWv3 gives a significantly older age (although overlapping with much larger uncertainty). Just be transparent and up-front about the limitations of the data. This is also why it's good to provide the code so anyone can see what's happening.

We have added text noting the large uncertainties of our dataset.  In this case, older ages would still imply ice loss prior to MWP-1a, however, so we retain the text comparing our results to published literature suggesting only modest East Antarctic contributions to MWP-1a.

Line 298:    Suggest starting with 'Our new in situ 14C results provide improved constraints…'

We modified this sentence similar to that suggested: "Our new in-situ $^{14}$C results provide improved constraints on past East Antarctic Ice Sheet thickness at a site ~400 km inland from the present-day coast".

Line 300:    Again, 'thicker' and 'thinner' are relative terms - say relative to what. Suggest just saying 'thicker than present at the LGM'

We have changed this text to "…thicker than at present at the LGM, but…".

Line 302:    'between thinner ice in the interior and thicker ice at the coast relative to today' is confusing to me as worded - suggest rewording to clarify that the hinge zone separates the thinner-than-present LGM ice in the interior from the thicker-than-present LGM ice at the coast.

We agreed that this wording is confusing and have changed the sentence to "*…the 'hinge zone' separating the interior ice (which was thinner at the LGM than it is today) from the ice nearer the coast (which was thicker at the LGM than it is today) was…*".

Line 306:    Please provide all relevant code used for the calculations in this paper.

SUPPLEMENT

Line 1:    Split Table S1 into at least two tables: 14C measurement data and ages and 10/26 measurement data and ages - all ages should be recalibrated from the original paper using UWv3 or publicly available code from the authors.

We split Table S1 into Tables S1 and Tables S4-6, with all ages recalibrated.

   For clarity I would suggest combining value and uncertainty columns (i.e., x.xxx±y.yyy) and have them at the same exponent and a common significant figure level (e.g., both should be 10^3 at/g or 10^4 at/g, not one at 10^3 and one at 10^4)

We have amended as suggested.

   Use the symbol for permil for the units in the stable C column

We made this change.

   Quartz column: GR01 should read 0.6034. Have all values in this column at 4 decimal places.

We set all values in this column to four decimal places.

   All values in Carbon yield and diluted carbon columns should have one decimal place

==We changed all C-yield values to one decimal place.==

    I would recommend 4 decimal places for all scientific notations, and combine value ± uncertainty in a single column with a single exponent common to the entire column. Pay attention to significant figures. Carry as many as possible through each column so the reader can arrive at the same value as the authors. But no need to carry extra, as in the 10/9 ratio column. No way we know those numbers better than the nearest 10-100e-15 values.

==We have elected to report each scientific-notation value in the supplement to the number of decimal places that limits the largest value in its column to four digits.  The Be and Al data come directly from previously published work, and we report it here exactly as it was originally reported, so that readers can arrive at the same value as the authors.==

Line 3:     'Table of sample measurement details' should be "Notes" below the table.

==We adjusted "Notes" below tables accordingly.==

    '0.58 ± 0.31' 1 sigma? 2 sigma? Standard deviation? Seems like something like standard deviation from the blank data in Balco et al. (2023). Present all blank data from the time of the extractions, or reference blanks presented in Balco et al (2023) for that period if that is a complete record of that time period. If that dataset is valid, the blank fluctuated by about a factor of 4 over the period covering the TUCNL numbers represented here.

==Our blank subtraction represents the mean and standard deviation of the number of atoms in the blanks run concurrently to our samples.  A note is added to the main text to clarify.==

    'Where the 1 sigma…': So, are all measurement uncertainties in this paper quoted as 1 sigma? At any rate the 6% value is on a concentration – that is not applicable to any of the other measurement columns. Clarify that. As noted earlier I'm also dubious about the 6% value since the CRONUS-A data in Balco et al. (2023) has a standard deviation of ca. 8.3%, and if you just look at the subset of measurements from Goehring et al. (2019), that value is ca. 8.7%. And the last two CRONUS-A measurements in Balco et al. (2023) are significantly higher than any of the previous values, and stop quite a bit earlier than the sample numbers here (60-100 samples later than the last of the ones listed in Balco et al.) – standard deviation of the whole dataset is over 10% if those are included. Include any additional CRONUS-A measurements from the time period spanning the measurements here to demonstrate either that the two high values are just scatter significantly outside the mean of the other samples, or that they represent a new mean if there was some sort of procedural change that happened to cause them to be higher from that point onward. In which case the default production rate in UWv3 is incorrect. If there was a procedural change at that time, that should also be clearly described and justified.

==The square brackets surrounding the "$^{14}C$" in that column heading indicate that those values are concentrations.  We adopt the 6% value to maintain continuity with previously published literature which used this value (e.g., Nichols *et al.*, 2019; Goehring *et al.*, 2019).==

Line 12:    If you need to break this 14C table across two pages you should just make a second table with 14C concentrations and ages, but also have a column on the left with the sample IDs for each page. Same for a separate table with 10Be and 26Al measurements – each should have the IDs.

We split these tables as suggested.

Blanks for the system need to be presented for the time frame of the extractions here. The authors need to demonstrate that they are consistent with what they claim for the long-term blank, which was calculated from earlier data. As noted earlier, in Balco et al. (2023) there were periods in which the blanks deviated from that mean value by quite a bit.

We have chosen to use a blank derived from the mean and standard deviation of the blanks run concurrently to our samples (Table S2).

Define 'effective blank'. This should be listed next to the blank-corrected total 14C columns. Also consider having just a % uncertainty column as with 10Be and 26Al.

Columns are renamed as with $^{10}$Be and $^{26}$Al.

Notes for this table should indicate how the ages were calculated, and which production rate dataset was used.

The production-rate dataset (that of Borchers *et al.*, 2016) is specified.

Line 21:    In general it is clearer in tables to have the units entirely below the column heading, in a different size or typeface (bold, italic, etc.), instead of running on to the end of the heading without any typographic differences.

We separate units by line breaks, and bold column headings.

The 27 in 27Al should be a superscript

"27" is now superscripted.

What sigma level is represented by the uncertainties? Are they treated similarly to the 14C, with comparison to replicate CRONUS-A or another repeat measurement?

Uncertainties in this data are taken directly from the original publications, which use no sigma notation.  We thus assume they are 1 sigma values.  Be concentrations are measured with respect to NIST SRM-4325 and Al concentrations SRM PRIME-289-0221, respectively.

The [26Al] % uncertainty column does not reflect the uncertainties and measurements in the previous column of atoms/g

This is a typo, which is fixed.

---

## Referee Report (RR1)

I agree with the authors' updates to this manuscript based on the reviewer comments, which I think helped clarify and strengthen the authors' arguments. This a really, really nice paper with important implications for EAIS volume during the LGM and subsequent thinning rates which will be useful targets for future glaciological modeling efforts. Thanks for a fun read!

I do have some very minor technical corrections – just a few places where clarity could be further improved and some typos. Most of my suggestions are for the abstract, as I think it could be tightened up a bit so that it's easy for a potential reader to hone in on the important findings here. I provided concrete wording suggestions but of course the authors' should feel free to use different wording that is more consistent with their voice! All this being said, I think this work is ready for publication.

**Abstract**

Line 10 – Make clear at the beginning that this study is in the Grove Mtns so you can more easily refer to it later in the abstract: "…14C dating [at a site/from a nunatak] in the Grove Mountains, located on the edge of the East Antarctic Plateau and 380 km inland…"

Line 11 – I think this second sentence here is an important concept to include in the intro (as you do), but is unnecessary in the abstract.

Line 16 – "location dividing thicker vs. thinner ice" -  I think the wording on Line 56 is slightly clearer, so you could update this to read "…the magnitude of these thickness changes and the transition point from thicker-than-present to thinner-than-present LGM ice are poorly constrained"

Lines 17-27 –

"Geological reconstructions" to "bedrock erosion" – I suggest removing most of this background into and combining it with a description of the work presented in this paper.

"Here," to "gradual ice sheet thinning began ~16 ka" could be tightened up/reorganized a bit to provide a clearer description of your findings.

Together, these lines could look something like:

"Here, we reconstruct East Antarctic Ice Sheet (EAIS) thickness changes since the LGM at a nunatak in the Grove Mountains using *in situ* 14C, which circumvents the common issue of long-lived nuclide inheritance that leads to inaccurate records of LGM ice thickness. Samples between 1912 m a.s.l. and the modern ice margin (~1825 m a.s.l) yield 14C ages of X-X. Samples at and above 1912 m a.s.l. have saturated 14C concentrations, implying exposure of the nunatak summit through the LGM. We therefore place the LGM ice surface in the Grove Mountains ~70 m higher than at present. The unsaturated samples below 1912 m a.s.l. indicate that gradual thinning began ~16 ka, with some (25-45%) post-LGM thinning recorded ~16-11 ka and most (55-75%) recorded during the Holocene. Ice sheet models…"

Note that I removed reviewer 3's suggestion to define saturation in the abstract – I'm not actually certain I agree that "saturation" is overly colloquial and I think the definition could go into the introduction instead (which I think you already have).

**Other technical corrections**

Line 94 – explicitly (but briefly) state here why this is it a key site?

Line 92 - "how far inland ice was thicker" – you could instead use the term "hinge zone," which you've defined already defined really nicely.

Lines 127-128 – maybe this is already stated above and I missed it, but could add the scatter in the 10Be and 26Al data as another line of evidence for inheritance.

Line 131 – rather than listing sample names in the text, could list them in Table 1.

Line 182 and Figure 2 caption – GR12 seems to be within uncertainty of (and actually, slightly younger than) GR01? I wonder if GR15 is really your only outlier? This is such a nice dataset!

Figure 2 – if not too much a hassle, could you add a $2^{nd}$ Y axis showing elevation relative to modern ice surface (i.e., GR12 would be ~0)? Also, a general comment that came up for me when looking at this figure – could point out somewhere more explicitly in the abstract that you narrowed the LGM ice thickness to a very small window (~70-85 m above present), which is amazing! I know you do in the discussion on lines 211-213, so it's also okay to just leave it there.

Line 206 – add "all of" before "our samples"

Line 209 – "at least not long or deeply enough" – a bit confusing still. How about: "not sufficiently thick as to override the summit for a significant duration, although saturation doesn't preclude a short period (<x kyr) of cover or cover by thin (<10 m) ice"

Line 216 – "Up to 18 m of thinning" - separate into two sentences with $2^{nd}$ being about glacial overshoot.

Line 239 – suggest rewording here and elsewhere from "covered shallowly" to "covered by thin ice" (see earlier suggestion)

Figure 3 caption – "contours show C14 conc"? Without the sample concentrations on here, I think you need to explain this slightly differently because I was left looking for 14C concentrations – "Burial-history contour plot for a sample at 1912 m a.s.l in the Grove Mountains. Modeled glacial histories start at 50 ka with one episode of burial under >10 m of ice (no 14C production during burial). 14C concentrations in the sample are saturated at the model start"

What is the "lesser end" of the saturation window – I think I asked that before but I still don't understand (closer to the grey zone, I guess? Could this maybe just be "The sample only remains saturated if burial durations longer than X kyr happened before the LGM"?).

"Sample GR21 plots off the bottom-left" – same as confusion about first sentence, above.

Line 265 – remind reader here what the previous LGM reconstructions based on longer lived nuclides suggest? Is lost track of this, although it looks like you come back to it below?

Line 269 – "that" should be "than"

Figure 4 – 2nd to last sentence probably unnecessary since you state this nicely already in the first sentence of the caption.

Line 308 – revise slightly for clarity "Our record suggests that ice in the Grove Mountains began thinning ~16 ka, ~2 kyr later than the [more coastal?] Prince Charles Mountains, though the timing of initial thinning at our site is broadly consistent with…."

Line 312 – CRONUS-Earth calculator sentence- this could go in methods. Just there state that all previously published ages presented here are recalculated using…

Line 314 – "Note, however" sentence could be – "Part of this discrepancy could be due to small amounts of 10Be and 26Al inheritance in the Prince Charles Mountain samples. Further work to measure *in situ* 14C in the Prince Charles Mountain samples would enable an evaluation of the degree of lead and lag…" (the last sentence of this paragraph isn't really necessary)

---

## Author Response (AR2)

I agree with the authors' updates to this manuscript based on the reviewer comments, which I think helped clarify and strengthen the authors' arguments. This a really, really nice paper with important implications for EAIS volume during the LGM and subsequent thinning rates which will be useful targets for future glaciological modeling efforts. Thanks for a fun read!

I do have some very minor technical corrections – just a few places where clarity could be further improved and some typos. Most of my suggestions are for the abstract, as I think it could be tightened up a bit so that it's easy for a potential reader to hone in on the important findings here. I provided concrete wording suggestions but of course the authors' should feel free to use different wording that is more consistent with their voice! All this being said, I think this work is ready for publication.

==Thank you – we appreciate your kind words and helpful suggestions!==

**Abstract**

Line 10 – Make clear at the beginning that this study is in the Grove Mtns so you can more easily refer to it later in the abstract: "…14C dating [at a site/from a nunatak] in the Grove Mountains, located on the edge of the East Antarctic Plateau and 380 km inland…"

==We have altered the text to read "In this study, we present a surface-exposure chronology of past ice-thickness change derived from *in-situ* cosmogenic-$^{14}$C dating at a site in the Grove Mountains, located on the edge of the East Antarctic plateau, 380 km inland from the coastline in the Lambert Glacier-Amery Ice Shelf sector.".==

Line 11 – I think this second sentence here is an important concept to include in the intro (as you do), but is unnecessary in the abstract.

==We have removed this sentence.==

Line 16 – "location dividing thicker vs. thinner ice" -  I think the wording on Line 56 is slightly clearer, so you could update this to read "…the magnitude of these thickness changes and the transition point from thicker-than-present to thinner-than-present LGM ice are poorly constrained"

==We have incorporated the suggested text.==

Lines 17-27 –

"Geological reconstructions" to "bedrock erosion" – I suggest removing most of this background into and combining it with a description of the work presented in this paper.

"Here," to "gradual ice sheet thinning began ~16 ka" could be tightened up/reorganized a bit to provide a clearer description of your findings.

Together, these lines could look something like:

"Here, we reconstruct East Antarctic Ice Sheet (EAIS) thickness changes since the LGM at a nunatak in the Grove Mountains using *in situ* 14C, which circumvents the common issue of long-lived nuclide inheritance that leads to inaccurate records of LGM ice thickness. Samples between 1912 m a.s.l. and the modern ice margin (~1825 m a.s.l) yield 14C ages of X-X.

Samples at and above 1912 m a.s.l. have saturated 14C concentrations, implying exposure of the nunatak summit through the LGM. We therefore place the LGM ice surface in the Grove Mountains ~70 m higher than at present. The unsaturated samples below 1912 m a.s.l. indicate that gradual thinning began ~16 ka, with some (25-45%) post-LGM thinning recorded ~16-11 ka and most (55-75%) recorded during the Holocene. Ice sheet models…"

Note that I removed reviewer 3's suggestion to define saturation in the abstract – I'm not actually certain I agree that "saturation" is overly colloquial and I think the definition could go into the introduction instead (which I think you already have).

We have changed the sentences on lines 15-23 to "However, the magnitude of these thickness changes and the transition point from thicker-than-present to thinner-than-present LGM ice are poorly constrained. Here, we reconstruct changes in the thickness of the East Antarctic Ice Sheet since the LGM at a nunatak in the Grove Mountains using *in-situ* [14], which circumvents the common issue of long-lived nuclide inheritance that leads to inaccurate records of LGM ice thickness. Samples between 1,912 m above sea level (a.s.l.) and the modern ice margin (~1,825 m a.s.l.) yield [14]C ages of 0.18

**Other technical corrections**

Line 94 – explicitly (but briefly) state here why this is it a key site?

We have added the following text to lines 87-90: "**The Grove Mountains are a key site for testing the location of this "hinge zone" as they lie close to the elevation of this feature identified in previous studies of the region (e.g., White *et al.*, 2011a).**".

Line 92 - "how far inland ice was thicker" – you could instead use the term "hinge zone," which you've defined already defined really nicely.

We have replaced the text in question with "the position of the "hinge zone" in this region".

Lines 127-128 – maybe this is already stated above and I missed it, but could add the scatter in the 10Be and 26Al data as another line of evidence for inheritance.

We have changed the sentence on lines 125-128 to read "As neither plucking scars nor glacial striae were observed at the site (Lilly *et al.*, 2010), indicating low or negligible rates of subglacial erosion, and because of the scatter observed in the [10]Be and [26]Al data from this site, we anticipate that the existing nuclide concentrations do not accurately record LGM ice thickness.".

Line 131 – rather than listing sample names in the text, could list them in Table 1.

We have removed the lists of sample names. We replaced the list on line 123 with "marked in **Table 1** with "(BR)" appended to the Sample ID".

Line 182 and Figure 2 caption – GR12 seems to be within uncertainty of (and actually, slightly younger than) GR01? I wonder if GR15 is really your only outlier? This is such a nice dataset!

We have changed the sentences on lines 183-185 to read "We suspect that the site of GR15 may have been covered by snow or other sediment, though we have not acquired any field evidence tot his effect. GR12 may instead not be an outlier, as its concentration lies within the uncertainty window of that of GR01.".

Figure 2 – if not too much a hassle, could you add a 2nd Y axis showing elevation relative to modern ice surface (i.e., GR12 would be ~0)? Also, a general comment that came up for me when looking at this figure – could point out somewhere more explicitly in the abstract that you narrowed the LGM ice thickness to a very small window (~70-85 m above present), which is amazing! I know you do in the discussion on lines 211-213, so it's also okay to just leave it there.

We have added second y-axes to parts (**b-d**) of this figure showing elevations relative to the modern ice surface (here taken to be only the ice surface measured during the second sampling campaign).

Line 206 – add "all of" before "our samples"

We have added the suggested text to line 211.

Line 209 – "at least not long or deeply enough" – a bit confusing still. How about: "not sufficiently thick as to override the summit for a significant duration, although saturation doesn't preclude a short period (<x kyr) of cover or cover by thin (<10 m) ice"

We have changed the sentence on lines 212-218 to read "These results thus show that ice was thicker at the LGM than at present in the Grove Mountains but not sufficiently thick as to override the summit for a significant duration, though saturation precludes neither a short period (<3 kyr) of cover nor cover by thin (<10 m) ice (**Fig. 3**)." And moved **Fig. 3** and its caption up to lines 216 and 217-230, respectively.

Line 216 – "Up to 18 m of thinning" - separate into two sentences with 2nd being about glacial overshoot.

We have changed this sentence to read "Up to 18 m (21-29%) of thinning could have occurred before and up to 21 m (24-33%) during meltwater pulse 1a (MWP-1a; **Fig. 2d**), ~13.5-14.7 ka, assuming the mean exposure ages of GR04 and GR18 and a linear thinning history. The potential for glacial overshoot, whereby the glacier thins beyond its new equilibrium thickness and subsequently rethickens, however, makes these minimum estimates.

Line 239 – suggest rewording here and elsewhere from "covered shallowly" to "covered by thin ice" (see earlier suggestion)

We have changed the sentence on lines 259-261 to read "The summit of the nunatak was either not covered or only covered briefly (≲1 kyr) or by thin (≲10 m) ice enough for the two summit samples to become re-saturated with $^{14}$C following re-exposure (**Fig. 3**)."

Figure 3 caption – "contours show C14 conc"? Without the sample concentrations on here, I think you need to explain this slightly differently because I was left looking for 14C concentrations – "Burial-history contour plot for a sample at 1912 m a.s.l in the Grove Mountains. Modeled glacial histories start at 50 ka with one episode of burial under >10 m of ice (no 14C production during burial). 14C concentrations in the sample are saturated at the model start"

What is the "lesser end" of the saturation window – I think I asked that before but I still don't understand (closer to the grey zone, I guess? Could this maybe just be "The sample only remains saturated if burial durations longer than X kyr happened before the LGM"?).

"Sample GR21 plots off the bottom-left" – same as confusion about first sentence, above.

We have altered the first sentences of the caption to read: "**Burial-history contour plot for a sample at 1,912 m above sea level (a.s.l.) in the Grove Mountains. Modeled glacial histories start at 50 ka with one episode of burial under >10 m of ice. No $^{14}$C is produced while buried. $^{14}$C concentrations in the sample are saturated at model start.**". We have changed "**lesser**" to "**lower-concentration**". The mean age of GR21 is technically oversaturated, so it will not appear on this plot, which asymptotes to the mean saturation concentration at its bottom-left corner. GR21 will thus always plot off the boundary of this graph.

Line 265 – remind reader here what the previous LGM reconstructions based on longer lived nuclides suggest? Is lost track of this, although it looks like you come back to it below?

We have changed the sentence on lines 265-270 to read "While we cannot rule out the ticker-than-present ice at the Grove Mountains being an entirely localized phenomenon, we suggest based on the application of $^{14}$C in this study and other Antarctic studies (e.g., White *et al.*, 2011b; Fogwill *et al.*, 2014; Nichols *et al.*, 2019; Hillenbrand *et al.*, 2021) that at least some previous reconstructions of LGM ice thickness based on longer-lived nuclides (e.g., $^{10}$Be and $^{26}$Al) which show either thickening or no thinning since the LGM away from the coast and fastest-flowing parts of East Antarctica may be inaccurate.".

Line 269 – "that" should be "than"

We have corrected this typo.

Figure 4 – 2nd to last sentence probably unnecessary since you state this nicely already in the first sentence of the caption.

The first sentence of the caption notes that the figure is modified from **Fig. 1c**, but not explicitly that the lines have been straightened. Another reviewer requested that we note explicitly why we straightened the lines in this depiction, so we would prefer to leave it in, repetitious though it may be.

Line 308 – revise slightly for clarity "Our record suggests that ice in the Grove Mountains began thinning ~16 ka, ~2 kyr later than the [more coastal?] Prince Charles Mountains, though the timing of initial thinning at our site is broadly consistent with…."

We have added the suggested text.

Line 312 – CRONUS-Earth calculator sentence- this could go in methods. Just there state that all previously published ages presented here are recalculated using…

We have edited this sentence to read "Cosmogenic-exposure ages reported here from other studies were recalculated using the online exposure age calculator formerly known as the

CRONUS-Earth online exposure age calculator [hess.ess.washington.edu], as with our $^{14}$C ages, using the primary $^{10}$Be calibration dataset of Borchers *et al.*, 2016.".

Line 314 – "Note, however" sentence could be – "Part of this discrepancy could be due to small amounts of 10Be and 26Al inheritance in the Prince Charles Mountain samples. Further work to measure *in situ* 14C in the Prince Charles Mountain samples would enable an evaluation of the degree of lead and lag…" (the last sentence of this paragraph isn't really necessary)

We have changed the sentences on lines 3190-323 to read "Part of this discrepancy could be due to inheritance in the samples from the PCMs. Further work to measure *in-situ* $^{14}$C in samples from the PCMs would enable an evaluation of the degree of lead and lag between sites.".